# USP16 counteracts mono-ubiquitination of RPS27a and promotes maturation of the 40S ribosomal subunit

Christian Montellese[1†], Jasmin van den Heuvel[1,2], Caroline Ashiono[1], Kerstin Dörner[1,2], André Melnik[3‡], Stefanie Jonas[1§], Ivo Zemp[1], Paola Picotti[3], Ludovic C Gillet[1], Ulrike Kutay[1]*

[1]Institute of Biochemistry, ETH Zurich, Zurich, Switzerland; [2]Molecular Life Sciences Ph.D. Program, Zurich, Switzerland; [3]Institute of Molecular Systems Biology, ETH Zurich, Zurich, Switzerland

**Abstract** Establishment of translational competence represents a decisive cytoplasmic step in the biogenesis of 40S ribosomal subunits. This involves final 18S rRNA processing and release of residual biogenesis factors, including the protein kinase RIOK1. To identify novel proteins promoting the final maturation of human 40S subunits, we characterized pre-ribosomal subunits trapped on RIOK1 by mass spectrometry, and identified the deubiquitinase USP16 among the captured factors. We demonstrate that USP16 constitutes a component of late cytoplasmic pre-40S subunits that promotes the removal of ubiquitin from an internal lysine of ribosomal protein RPS27a/eS31. *USP16* deletion leads to late 40S subunit maturation defects, manifesting in incomplete processing of 18S rRNA and retarded recycling of late-acting ribosome biogenesis factors, revealing an unexpected contribution of USP16 to the ultimate step of 40S synthesis. Finally, ubiquitination of RPS27a appears to depend on active translation, pointing at a potential connection between 40S maturation and protein synthesis.

**\*For correspondence:**
ulrike.kutay@bc.biol.ethz.ch

**Present address:** [†]CSL Behring, CSL Biologics Research Center, Bern, Switzerland; [‡]MSD Merck Sharp & Dohme AG, Lucerne, Switzerland; [§]Institute of Molecular Biology and Biophysics, ETH Zurich, Zurich, Switzerland

**Competing interests:** The authors declare that no competing interests exist.

## Introduction

Ribosomes stand at the center of translation in all kingdoms of life, catalyzing the synthesis of proteins by reading a messenger RNA (mRNA) template. Since translation is key to cellular growth and proliferation, the production of ribosomes is tightly regulated. Ribosome biogenesis encompasses the action of more than 200 non-ribosomal proteins, so-called assembly or trans-acting factors that collectively ensure that the two ribosomal subunits, a small 40S and a large 60S subunit in eukaryotes, are correctly assembled.

While the initial steps of eukaryotic ribosome synthesis leading to the formation of immature pre-ribosomal subunits occur in the nucleolus, ribosomal subunits gain competence for protein translation only during their ultimate maturation in the cytoplasm. This involves structural rearrangements of the subunits, incorporation of late-assembling ribosomal proteins, final ribosomal RNA (rRNA) processing steps, and the eventual release of all ribosome assembly factors (reviewed in *Cerezo et al., 2019*; *Chaker-Margot, 2018*; *Nerurkar et al., 2015*; *Peña et al., 2017*; *Woolford and Baserga, 2013*; *Zemp and Kutay, 2007*). Early cytoplasmic pre-40S particles contain a similar set of at least eight trans-acting factors in both yeast and human cells, namely ENP1, LTV1, casein kinase 1/Hrr25, RRP12, TSR1, the protein kinase and ATPase RIOK2, the endonuclease NOB1, and DIM2/PNO1 (partner of NOB1) (*Ameismeier et al., 2018*; *Ferreira-Cerca et al., 2012*; *Ghalei et al., 2015*; *Granneman et al., 2010*; *Hector et al., 2014*; *Heuer et al., 2017*; *Johnson et al., 2017*; *Larburu et al., 2016*; *Merl et al., 2010*; *Mitterer et al., 2019*; *Oeffinger et al., 2004*; *Scaiola et al., 2018*; *Schäfer et al., 2006*; *Schäfer et al., 2003*;

*Strunk et al., 2011*; *Wyler et al., 2011*; *Zemp et al., 2014*; *Zemp et al., 2009*). The release of these factors is mechanistically linked to defined steps of subunit maturation during which ribosomal proteins and some RNA segments gain their final position. Only few trans-acting factors have been found associated with pre-ribosomal subunits just prior to the final processing of the 3' end of 18S rRNA (*Fatica et al., 2003*; *Pertschy et al., 2009*; *Preti et al., 2013*; *Sloan et al., 2013*). In addition to the endonuclease NOB1, these include DIM2 and the protein kinase RIOK1 (*Ameismeier et al., 2018*; *Hector et al., 2014*; *Turowski et al., 2014*; *Widmann et al., 2012*). The ATPase activity of RIOK1 is required for rRNA cleavage and the eventual dissociation of NOB1 and DIM2 from the subunit (*Ferreira-Cerca et al., 2014*; *Turowski et al., 2014*; *Vanrobays et al., 2001*; *Widmann et al., 2012*). Based on recent structural studies, release or repositioning of DIM2 has been suggested to be necessary for both rRNA cleavage and formation of the 40S subunit decoding site (*Ameismeier et al., 2018*). How exactly these steps are mechanistically linked remains, however, ill-defined.

These final stages of the ribosome assembly pathway take place in the cytoplasm in presence of the entire translation machinery including mRNAs, tRNAs, the mature partner subunits and translation factors. In light of this fact, trans-acting factors may not only promote subunit maturation but also prevent an untimely engagement of pre-ribosomal subunits in translation. Although it is generally assumed that immature 40S subunits cannot engage in translation initiation, several studies have shown that pre-40S subunits can be detected in polysome fractions (*Belhabich-Baumas et al., 2017*; *Ferreira-Cerca et al., 2005*; *García-Gómez et al., 2014*; *Granneman et al., 2005*; *Lacombe et al., 2009*; *Pertschy et al., 2009*; *Soudet et al., 2010*). Whether these pre-40S particles are aberrant and therefore retain 40S biogenesis factors and/or whether entering the translating ribosome pool serves a surveillance function remains unclear. Final cytoplasmic pre-40S subunit maturation in *Saccharomyces cerevisiae* has been suggested to involve proofreading of functional sites (*Karbstein, 2013*; *Lebaron et al., 2012*; *Strunk et al., 2012*). In a so-called 'translation-like cycle', pre-40S particles have been shown to associate with mature 60S subunits generating a so-called 80S-like particle, which might be part of a final proofreading mechanism for 40S subunits (*Ferreira-Cerca et al., 2014*; *García-Gómez et al., 2014*; *Ghalei et al., 2017*; *Turowski et al., 2014*). Yet, it remains unclear whether 60S subunit association presents an obligatory step in final 40S subunit maturation or if only a subset of 40S precursors undergoes this quality control process (*Cerezo et al., 2019*; *Lebaron et al., 2012*; *Strunk et al., 2011*; *Strunk et al., 2012*; *Turowski et al., 2014*). While cytoplasmic events of 40S subunit maturation in human and yeast are presumed to be highly similar with few functional differences (*Badertscher et al., 2015*; *Carron et al., 2011*; *Wild et al., 2010*; *Wyler et al., 2011*; *Zorbas et al., 2015*), the existence of an 80S-like particle involved in pre-40S proofreading has not been described in human cells. It therefore remains unclear whether the transition from a pre-40S particle to a mature 40S subunit is assisted by additional, so far unidentified factors besides mature 60S subunits or only requires the well-described set of 40S trans-acting factors involved in cytoplasmic pre-40S maturation. Such additional factors might have been missed so far due to a sub-stoichiometric or transient mode of action.

To discover novel factors involved in the last stages of pre-40S subunit maturation, we isolated late 40S precursors from human cells and identified associated proteins by mass spectrometry. This led to the identification of the deubiquitinase USP16, previously shown to be involved in the deubiquitination of histone H2A (*Cai et al., 1999*; *Joo et al., 2007*). Our analysis revealed that USP16 is a cytoplasmic protein that possesses a novel, ribosome-associated function. We demonstrate that USP16 is a component of late cytoplasmic pre-40S particles and that its deletion affects the last stages of 40S maturation. Further, we show that loss of USP16 leads to the accumulation of mono-ubiquitinated 40S ribosomal protein RPS27a/eS31. Finally, our data suggest that RPS27a ubiquitination depends on active translation, hinting at a potential link between the final stages of 40S maturation and translation.

## Results

### Identification of USP16 as a pre-40S-associated factor

During the final step of 40S ribosomal subunit maturation in the cytoplasm, the 18S-E pre-rRNA is processed to its mature form by the endonuclease NOB1 (*Preti et al., 2013*; *Sloan et al., 2013*). At

this stage, pre-40S particles are known to contain only a few additional trans-acting factors including the NOB1 binding partner DIM2 and the atypical kinase RIOK1 (*Ameismeier et al., 2018*; *Hector et al., 2014*; *Turowski et al., 2014*; *Widmann et al., 2012*). While DIM2 and NOB1 are associated with pre-40S subunits already during earlier nuclear and cytoplasmic 40S subunit maturation steps (*Ameismeier et al., 2018*; *Larburu et al., 2016*; *Montellese et al., 2017*; *Wyler et al., 2011*; *Zemp et al., 2014*; *Zemp et al., 2009*), RIOK1 associates with pre-40S particles only in the course of their final maturation (*Ameismeier et al., 2018*; *Widmann et al., 2012*). However, additional as of yet unknown factors might be needed to control or support the final 40S ribosomal subunit biogenesis step.

To identify human 40S ribosomal subunit biogenesis factors associated with late cytoplasmic 40S subunit precursors, we isolated pre-40S particles by StrepTactin affinity purification of the C-terminally Strep-HA (StHA)-tagged kinase RIOK1 that we expressed from a tetracycline-inducible transgene in HEK293 cells (*Figure 1A*). As the kinase-dead (kd) D324A mutant of RIOK1 has been demonstrated to be more strongly associated with pre-40S particles than the wild-type (WT) kinase (*Widmann et al., 2012*), we also used RIOK1(kd)-StHA as bait. Affinity purifications were performed in biological triplicates and the respective eluates were analyzed by mass spectrometry using data-dependent acquisition (*Supplementary file 2*). As RIOK1 is also part of the methylosome that methylates Sm proteins during snRNP biogenesis (*Guderian et al., 2011*), the methylosome components PRMT5 and MEP50 were strongly enriched on RIOK1 (*Figure 1A and B*, *Supplementary file 2*), as previously reported (*Widmann et al., 2012*). While the methylosome complex was enriched on WT and mutant RIOK1 to a similar extent, 40S ribosomal proteins (RPs) and the known 40S trans-acting factors DIM2, NOB1 and TSR1 were preferentially isolated with RIOK1(kd) particles (*Figure 1C*). TSR1 is a GTPase-like protein that binds to the subunit interface of pre-40S particles, where it is suggested to impede binding of certain translation factors (*Ameismeier et al., 2018*; *Carron et al., 2011*; *McCaughan et al., 2016*; *Mitterer et al., 2019*). In addition to ribosomal proteins and known 40S trans-acting factors, a number of proteins not previously linked to 40S subunit maturation were found to be strongly enriched on the kinase-dead version of RIOK1, with OTUD6B, G3BP1, and USP16 being the most significant among the top scorers (*Figure 1C*, *Supplementary file 2*). OTUD6B is a deubiquitinase (DUB) suggested to associate with and deubiquitinate components of the 48S translation pre-initiation complex, thereby regulating protein synthesis (*Sobol et al., 2017*). G3BP1 is a core component of stress granules and has been linked to diverse aspects of mRNA metabolism (reviewed in *Alam and Kennedy, 2019*). Lastly, USP16 has originally been described as a DUB of histone H2A and subsequently been implicated in the regulation of gene expression, cell cycle progression, differentiation, and the DNA damage response (*Cai et al., 1999*; *Gu et al., 2016*; *Joo et al., 2007*; *Yang et al., 2014*; *Zhang et al., 2014*). Interestingly, the increased gene dosage of *USP16*, which is located on human chromosome 21, has been linked to the manifestation of Down's syndrome (*Adorno et al., 2013*), whereas virus-induced downregulation of USP16 has been suggested to be a critical step in tumorigenicity of Hepatitis B virus (*Qian et al., 2016*). In this study, we decided to focus on USP16 to investigate its potential function in 40S subunit maturation.

First, to verify the results obtained by mass spectrometry, we repeated the RIOK1-StHA StrepTactin affinity purification and analyzed the eluates by immunoblotting (*Figure 1D*). This showed a significant enrichment of USP16 as well as NOB1 and the 40S ribosomal protein RPS3/uS3 on the RIOK1(kd)-StHA bait, confirming our MS analysis. To investigate whether USP16 can also be co-purified with other (pre-)40S associated factors, we next used RPS2/uS5 and the 40S trans-acting factors ENP1 and LTV1 as baits in comparison to RIOK1(kd) (*Figure 1E*). Immunoblotting revealed that USP16 was enriched in all purified 40S particles to an extent comparable to NOB1, but was absent in a pull-down of HASt-tagged GFP used as negative control. Finally, we determined the sedimentation behavior of USP16 in sucrose gradients of HEK293 cell extract (*Figure 1F*). Consistent with the assumption that USP16 is associated with pre-40S particles, USP16 co-migrated with LTV1, NOB1 and RPS3/uS3 in the 40S peak fractions but was enriched neither in the 60S peak nor in heavier fractions.

## USP16 is associated with late cytoplasmic 40S pre-ribosomes

To further characterize USP16 and its association with 40S-sized ribosomal particles, we tagged both WT USP16 and a catalytically inactive mutant, in which the active site cysteine at position 205 was mutated to serine (C205S; *Cai et al., 1999*), with a C-terminal StHA tag. First, to determine their

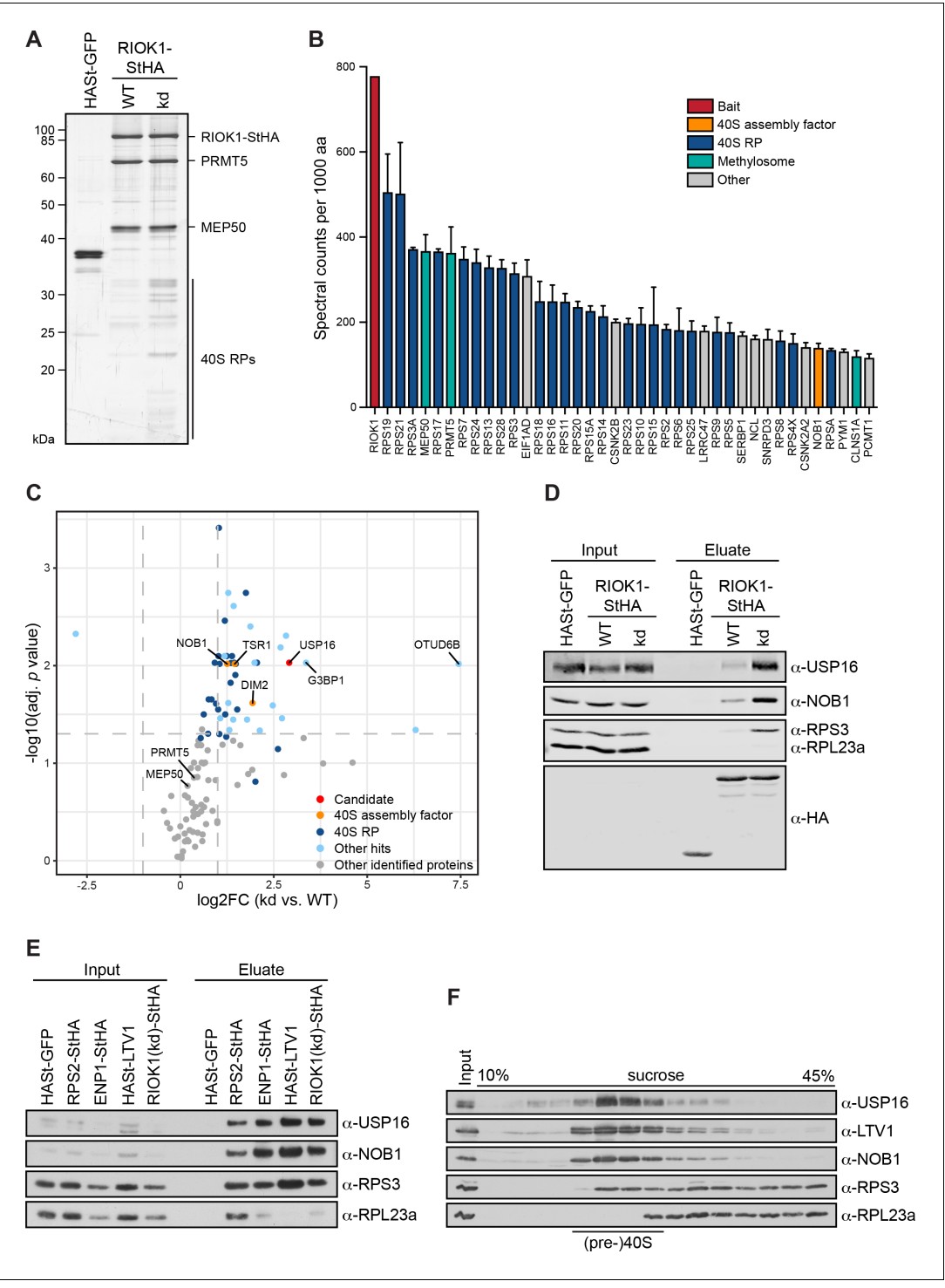

**Figure 1.** USP16 is a pre-40S associated factor. (**A**) StrepTactin affinity purification of HASt-GFP, wild-type (WT) and kinase-dead (kd) RIOK1-StHA from HEK293 cell lysates. Eluates were analyzed by SDS-PAGE and silver staining or mass spectrometry. (**B**) Proteomic analysis of three independent RIOK1(kd)-StHA StrepTactin affinity purifications. The plot shows spectral counts (mean ± SD) of the 40 top identified hits after filtering against the HASt-GFP control and after normalization to spectral counts of the bait protein as well as to the size of the protein in amino acids (aa). Data before and after normalization are shown in **Supplementary file 2**. (**C**) Proteomic analysis of three independent experiments as in (**A**). The plot shows the $\log_2$ fold change ($\log_2$FC) of the average number of spectral counts of proteins identified in the RIOK1(kd)-StHA versus the RIOK1(WT)-StHA pull down against the negative $\log_{10}$ of the adjusted $p$ value. Proteins with a fold change <2 and an adjusted $p$ value > 0.05, demarcated

*Figure 1 continued on next page*

*Figure 1 continued*

by dashed lines, were considered nonsignificant. (**D**) StrepTactin pull-downs of HASt-GFP, RIOK1(WT)- and RIOK1(kd)-StHA from HEK293 cell lysates. Eluates were analyzed by immunoblotting using the indicated antibodies. Load corresponds to 0.05% of the input and 20% of the eluates. (**E**) StrepTactin pull-downs of HASt-GFP, RPS2-StHA, ENP1-StHA, HASt-LTV1, and RIOK1(kd)-StHA from HEK293 cell lysates. Eluates were analyzed by immunoblotting using the indicated antibodies. Load corresponds to 0.05% of the input and 20% of the eluates. (**F**) HEK293 cell extract was separated on a linear 10–45% sucrose gradient by centrifugation. Input and gradient fractions were analyzed by immunoblotting using the indicated antibodies.

subcellular localization, we transfected these constructs into HeLa cells and observed that both USP16(WT)- and USP16(C205S)-StHA localized to the cytoplasm (*Figure 2A*) as described previously (*Cai et al., 1999*; *Zhang et al., 2014*). Several pre-40S-associated factors such as ENP1, DIM2, LTV1, and RIOK2 accompany immature 40S subunits from the nucleus to the cytoplasm. Since pre-40S export is dependent on the export factor CRM1/XPO1, they accumulate in the nucleoplasm upon its inhibition by the drug leptomycin B (LMB; *Rouquette et al., 2005*; *Thomas and Kutay, 2003*; *Zemp et al., 2014*; *Zemp et al., 2009*). In contrast, USP16-StHA did not accumulate in the nucleus upon treatment of cells with LMB (*Figure 2A*). Since USP16 is localized to the cytoplasm at steady-state and seems not to shuttle into the nucleus, we conclude that USP16 must be associated with a cytoplasmic but not a nuclear pool of (pre-)40S particles.

Next, we generated tetracycline-inducible HEK293 cell lines for expression of USP16(WT)- or USP16(C205S)-StHA and performed a one-step StrepTactin affinity purification in biological triplicates followed by mass spectrometry using data-dependent acquisition to analyze co-purifying proteins (*Figure 2B* and *Supplementary file 3*). Interestingly, among the main hits we identified many 40S RPs along with characterized 40S trans-acting factors known to be associated with cytoplasmic pre-40S subunits such as ENP1, LTV1, TSR1, DIM2, and NOB1. In contrast, only few 40S trans-acting factors known to associate with nuclear pre-40S particles (e.g. NOC4L, NOP14) or 60S trans-acting factors (e.g. PES1, NOG2) were identified and these were co-purified much less efficiently (*Figure 2C*, *Supplementary file 3*). These data confirm that USP16 is specifically associated with a cytoplasmic pool of pre-40S subunits.

To verify the data obtained from mass spectrometry, we performed tandem affinity purification (TAP) of WT and mutant USP16-StHA and analyzed the eluate by silver staining and immunoblotting (*Figure 2D and E*). This confirmed that pre-40S subunits were co-purified with both WT and mutant USP16-StHA as indicated by the presence of the late 40S trans-acting factor NOB1 and the 40S RP RPS3/uS3, whereas the early, nuclear 40S trans-acting factor NOC4L was not detected. The mutant form of USP16 was retrieved slightly less efficiently, which could explain the lower levels of 40S-associated proteins. The absence of the 60S trans-acting factor RLP24 and the ribosomal protein RPL23a/uL23 further confirmed that USP16 specifically binds to 40S precursors. Notably, the small ribosomal subunit protein RPS10/eS10 was also detected in the eluate. RPS10 has previously been suggested to stably join pre-40S subunits at a very late cytoplasmic step of 40S maturation, since it is not detected in earlier pre-40S particles purified by TAP of LTV1 or C21ORF70, a nuclear 40S assembly factor, or by Rio2 TAP from yeast (*Collins et al., 2018*; *Larburu et al., 2016*; *Strunk et al., 2011*; *Zemp et al., 2014*). The presence of RPS10 in USP16-associated 40S particles thus indicates that USP16 is associated with particles corresponding to pre-40S subunits at a very late stage of cytoplasmic maturation.

## Pre-40S association of USP16 depends on its ZnF-UBP domain and a USP16-specific helix in the USP domain

USP16 is a DUB of the ubiquitin-specific protease (USP) family, which comprises 54 members in humans (*Mevissen and Komander, 2017*). All USPs are characterized by a USP domain, which contains two highly conserved cysteine and histidine boxes that confer catalytic activity. The USP domain can differ considerably in length due to the insertion of polypeptide stretches (reviewed in *Bonnet et al., 2008*; *Komander et al., 2009*; *Nijman et al., 2005*). Further, USP-containing DUBs can have additional domains which contribute to ubiquitin (Ub) recognition or are involved in the regulation of DUB activity. USP16, in addition to its USP domain, contains an N-terminal zinc-finger ubiquitin binding domain (ZnF-UBP) domain (*Figure 3A*). The USP domain itself contains an insertion

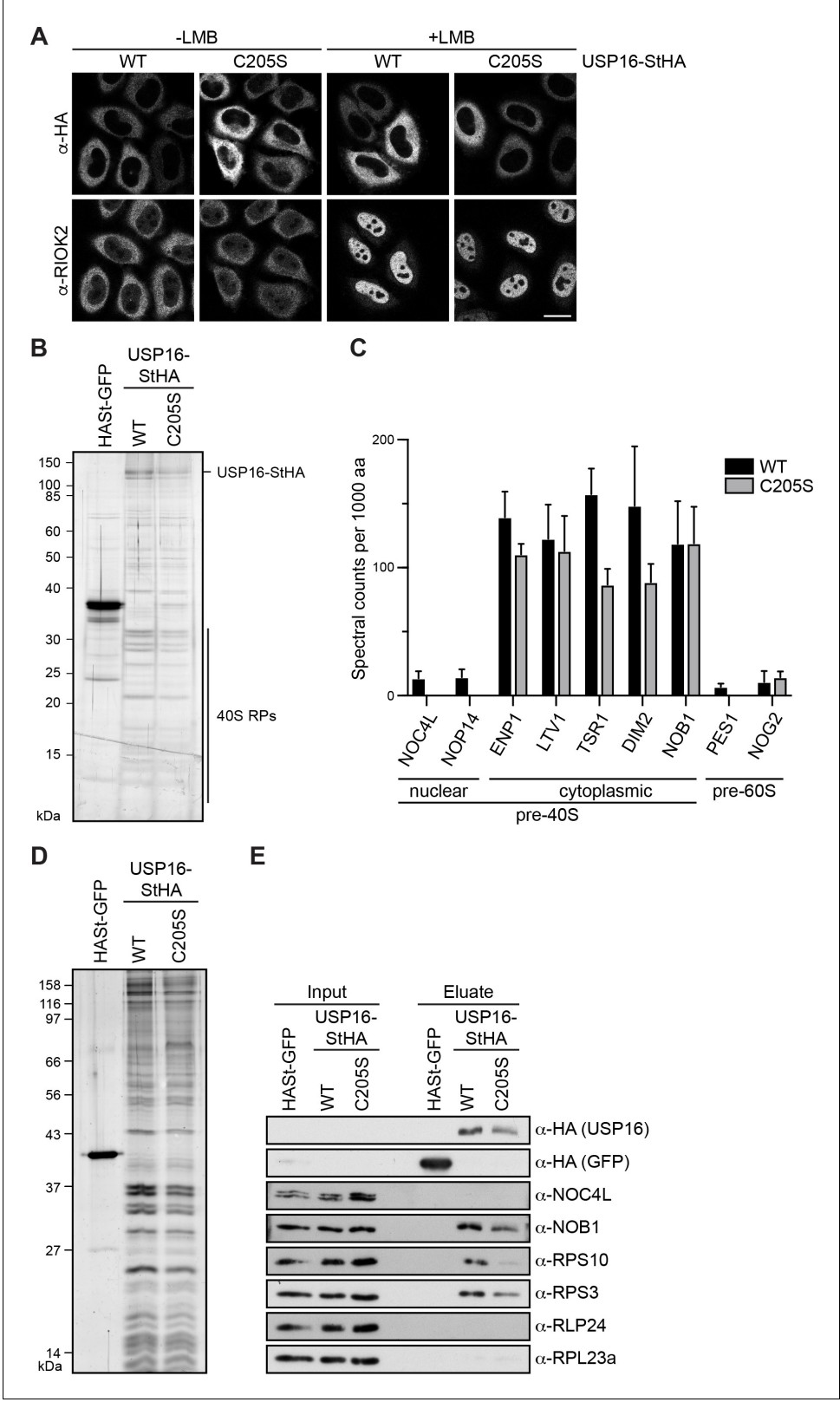

**Figure 2.** USP16 is associated with cytoplasmic pre-40S subunits. (**A**) USP16(WT)- and USP16(C205S)-StHA constructs were transiently transfected into HeLa cells and their localization was analyzed by immunostaining using an HA antibody 24 hr after transfection. CRM1-mediated nuclear export was inhibited by treatment with leptomycin B (LMB; 20 nM, 120 min) as indicated. Immunostaining of the 40S trans-acting factor RIOK2 served as

*Figure 2 continued on next page*

*Figure 2 continued*

positive control for the LMB treatment. Scale bar, 20 μm. (B) One-step StrepTactin affinity purification of HASt-GFP, wild-type (WT) and catalytically inactive (C205S) USP16-StHA from HEK293 cell lysates. Eluates were analyzed by SDS-PAGE and silver staining or mass spectrometry. (C) Proteomic analysis of three independent experiments as in (B). The plot shows spectral counts (mean ± SD) of identified candidate proteins normalized to spectral counts of the bait protein as well as to the size of the protein in amino acids (aa). Candidate proteins are categorized into 40S and 60S trans-acting factors. (D) Tandem affinity purification (TAP) from HEK293 cells expressing HASt-GFP, USP16(WT)- or USP16(C205S)-StHA. Eluates were analyzed by SDS-PAGE and silver staining. (E) Immunoblot analysis of (D) using the indicated antibodies against the bait (α-HA), 40S (NOB1, NOC4L) and 60S (RLP24) trans-acting factors, and RPs. Load corresponds to 0.05% of the input and to 20% of the eluates.

of approximately 200 amino acids (aa), which is not found in other members of the USP family except for USP45, the most closely related DUB of USP16 in vertebrates (*Clague et al., 2013*). USP45 seems, however, not to be expressed in various human cell lines (*Geiger et al., 2012*; *Nagaraj et al., 2011*). Based on secondary structure predictions, this insertion of 200 aa is mostly unstructured but contains a positively charged helix (aa 436–460), which could potentially contribute to binding of negatively charged rRNA.

To elucidate which domains of USP16 are required for 40S binding, we generated C-terminally StHA-tagged USP16 truncation constructs, which contained either only the ZnF-UBP domain (ZnF: aa 1–192) or the USP domain (USP: aa 193–822) (*Figure 3A*). Additionally, we deleted the basic helix in either full length USP16 (ΔH: aa 1-822Δ436–460) or in the isolated USP domain (USPΔH: aa 193-822Δ436–460) to examine a potential contribution of the helix to 40S binding. First, we analyzed the localization of these truncated constructs after transient transfection into HeLa cells (*Figure 3B*). The USP, ΔH and USPΔH fragments all localized to the cytoplasm like WT USP16-StHA. The ZnF domain was predominantly found in the cytoplasm but was also present in the nucleus, possibly due to its small size (28 kDa) allowing free diffusion between the nucleus and the cytoplasm.

To directly address which parts of USP16 contribute to 40S binding, we performed TAP from HEK293 cells inducibly expressing the mutant constructs and analyzed the co-purified proteins by silver staining and immunoblotting (*Figure 3C and D*). The isolated ZnF domain associated with pre-40S particles, although their enrichment was slightly diminished compared to the WT, especially considering the higher expression levels of the ZnF domain. In contrast, binding of pre-40S subunits to the USP domain was strongly compromised and only very low levels were detectable both by silver staining and immunoblotting. Further, deletion of the USP16-specific basic helix (ΔH) did not alter binding to pre-40S subunits compared to full-length USP16. However, the weak 40S association of the isolated USP domain was completely lost upon deletion of the basic helix. Taken together, we conclude that mainly the ZnF domain but also the positively charged helix within the USP domain contribute to pre-40S binding.

## Loss of USP16 leads to accumulation of modified RPS27a

Having established that USP16 is bound to pre-40S subunits, we sought to gain insights into a potential ribosome-associated function, which could entail deubiquitination of an unidentified USP16 substrate. Several RPs of the 40S subunit have been linked to ubiquitination and are thus candidate USP16 substrates. For instance, RPS2/uS5, RPS3/uS3, RPS3a/eS1, RPS10/eS10, and RPS20/uS10 contain sites for regulatory mono-ubiquitination (*Garzia et al., 2017*; *Higgins et al., 2015*; *Juszkiewicz and Hegde, 2017*; *Matsuo et al., 2017*; *Sundaramoorthy et al., 2017*; *Figure 4A*). Furthermore, the ribosomal protein RPS27a/eS31 is produced as a linear fusion with ubiquitin (Ub), which is cleaved off before the ribosome reaches translation competence (*Rabl et al., 2011*). To assess whether USP16 might be involved in the removal of Ub from these candidate RPs, we assessed their modification by immunoblotting upon depletion of USP16 by RNAi (*Figure 4B and C*). Interestingly, knockdown of USP16 with two different siRNAs led to the appearance of a higher molecular weight band of RPS27a, but not of the other ribosomal proteins tested. To verify the specificity of the USP16 knockdown, we performed RNAi rescue experiments in HEK293 cell lines inducibly expressing WT or catalytically inactive USP16-StHA using an siRNA targeting the 3'UTR of the USP16 mRNA (*Figure 4D*). As observed before, depletion of USP16 in either parental HEK293 cells or in uninduced USP16 cell lines led to the appearance of a higher molecular weight band of

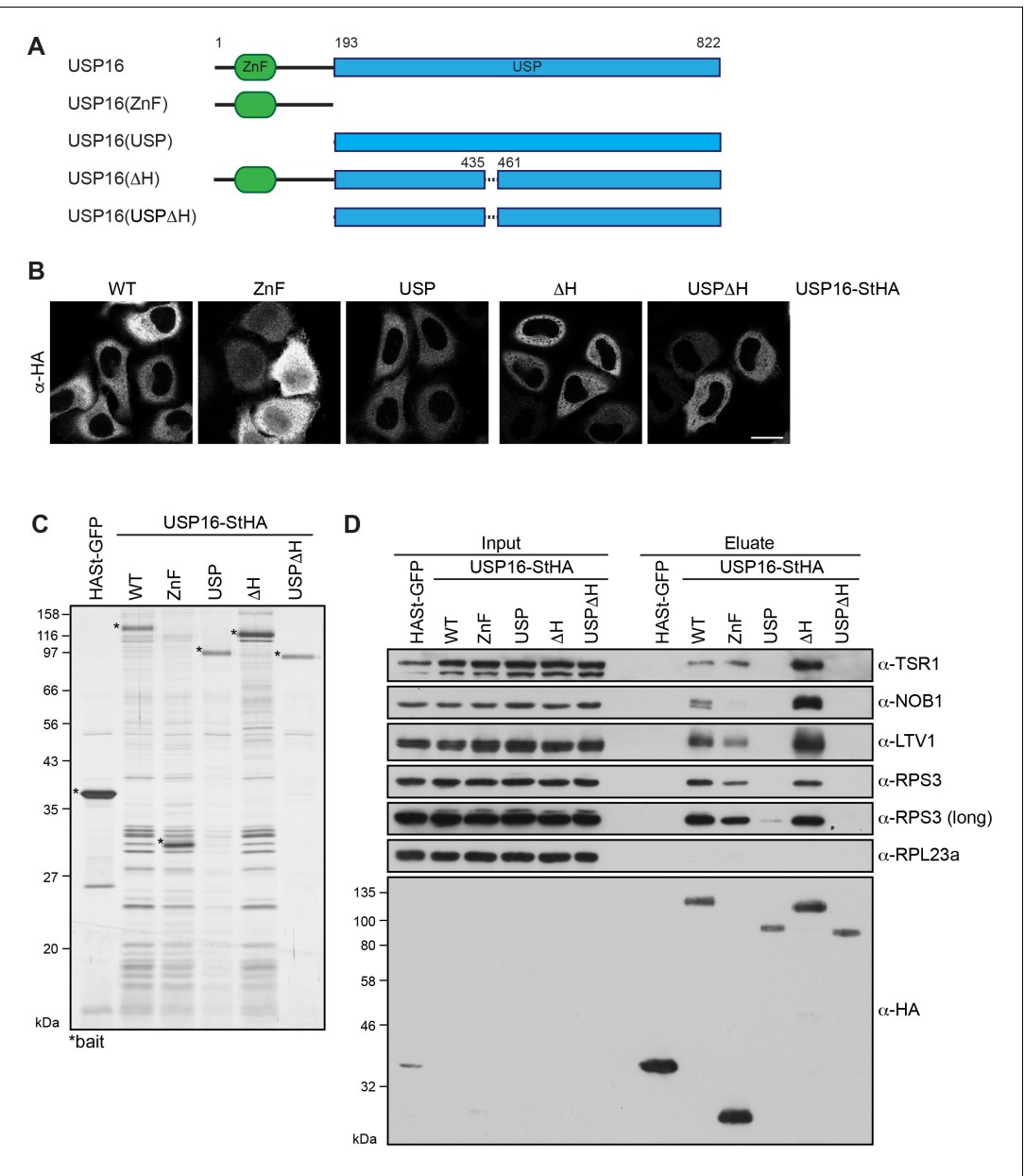

**Figure 3.** Pre-40S association of USP16 depends on its ZnF-UBP domain and a USP16-specific insertion in the USP domain. (A) Scheme of protein domains present in human USP16 and generated deletion constructs. USP16 contains an N-terminal zinc-finger ubiquitin binding domain (ZnF-UBP) domain and a C-terminal ubiquitin-specific protease (USP) domain. The amino acid numbers indicate the positions of the truncations, dashed lines represent deletions. (B) C-terminally StHA-tagged USP16 constructs, depicted in (A), were transiently transfected into HeLa cells and their localization was analyzed by immunostaining using an HA antibody 24 hr after transfection. Scale bar, 20 μm. (C) TAP from HEK293 cell lines expressing HASt-GFP or the USP16 constructs depicted in (A). Eluates were analyzed by SDS-PAGE and silver staining. Bait proteins are marked with an asterisk (*). (D) Immunoblot analysis of experiment in (C) using the indicated antibodies against 40S trans-acting factors, ribosomal proteins or the bait (α-HA). Two different exposures are shown for the RPS3 immunoblot. Load corresponds to 0.05% of the input and to 20% of the eluates.

RPS27a. Importantly, this could be rescued by tetracycline-induced expression of tagged WT but not mutant USP16, indicating that the catalytic activity of USP16 is required for removal of the RPS27a modification. Interestingly, StrepTactin affinity purification of mutant USP16 co-purified modified RPS27a even in presence of endogenous USP16 (*Figure 4E*), suggesting that catalytically

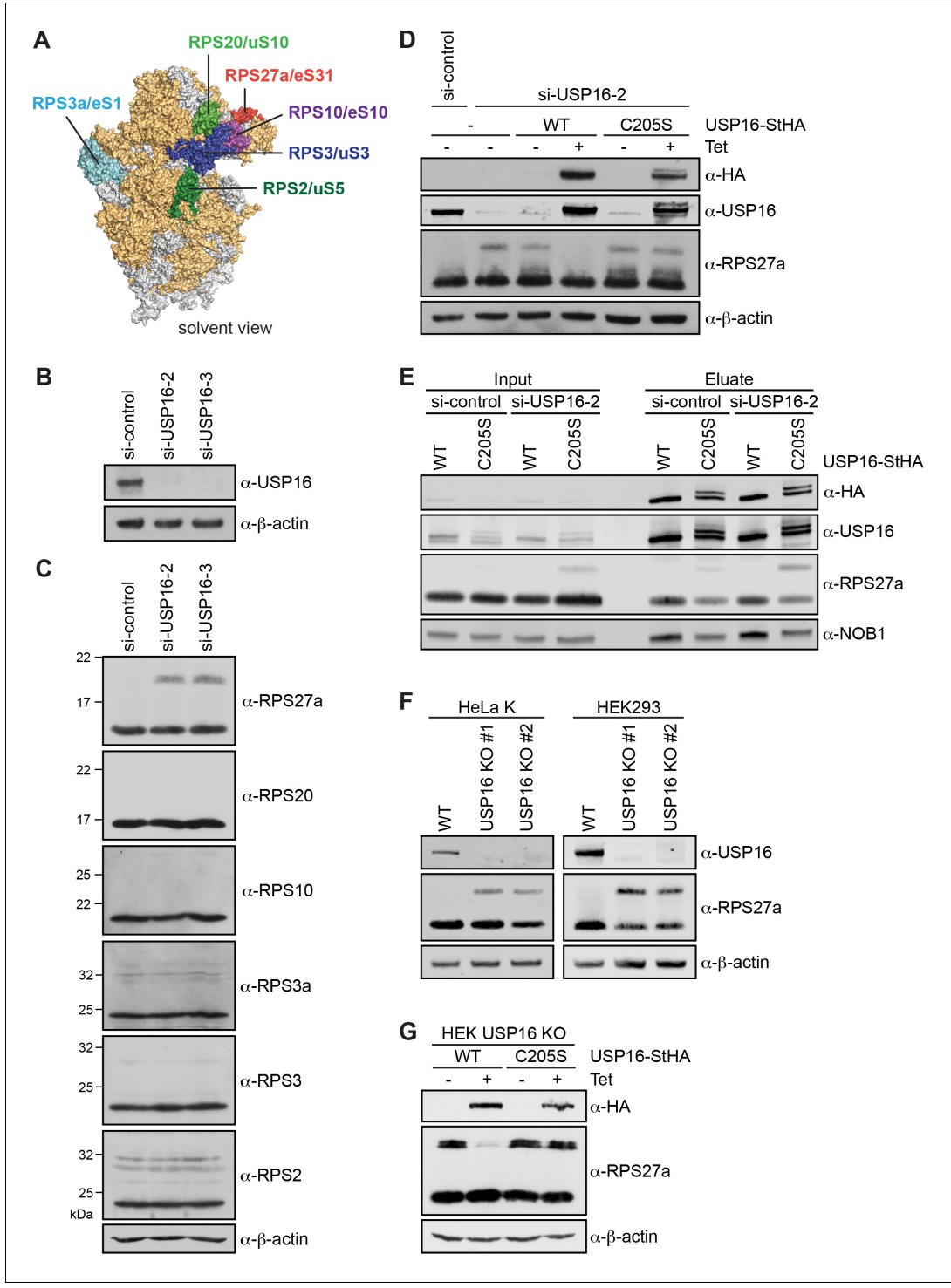

**Figure 4.** Depletion of USP16 leads to accumulation of modified RPS27a. (**A**) Structural model of the human 40S subunit shown from the solvent side (adapted from *Khatter et al., 2015*, PDB ID: 4UG0). Ribosomal proteins analyzed for mono-ubiquitination are highlighted, others RPs are shown in orange and 18S rRNA is shown in gray. (**B**) HeLa cells were treated with either control siRNA (si-control) or two different siRNAs against USP16 for 72 hr. To control for downregulation of USP16, cell extracts were analyzed by immunoblotting using a USP16 antibody. (**C**) Immunoblot analysis of cell extracts from (**A**) using the indicated antibodies against RPs of the 40S subunit. (**D**) Parental HEK293 cells or HEK293 cells expressing USP16(WT)- or USP16(C205S)-StHA were treated with a control siRNA or an siRNA targeting the 3'UTR of the USP16 mRNA for 72 hr. Cells expressing USP16(WT)- or USP16 (C205S)-StHA were either left uninduced (-Tet) or induced (+Tet) with tetracycline (0.5 µg/ml) for the last 48 hr of

*Figure 4 continued on next page*

*Figure 4 continued*

the siRNA treatment. Cell extracts were analyzed by immunoblotting using the indicated antibodies. (**E**) StrepTactin pull-down from lysates of HEK293 cells that had been transfected with either a control or a USP16 siRNA 72 hr before harvest and induced to express either USP16(WT)- or USP16(C205S)-StHA (0.5 μg/ml tetracycline) for the last 48 hr of the siRNA treatment. Inputs and eluates were analyzed by immunoblotting using the indicated antibodies. Note that the USP16(C205S)-StHA protein is potentially modified as it runs as two bands in SDS-PAGE. Load corresponds to 0.05% of the input and to 20% of the eluates. (**F**) USP16 knockout (KO) cell lines were generated using the CRISPR/Cas9 system in HeLa and HEK293 backgrounds using two different guide RNAs targeting exon 3 (KO #1) or exon 5 (KO #2) of USP16, respectively. Cell extracts were analyzed by immunoblotting using the indicated antibodies. (**G**) HEK293 *USP16* KO cells expressing USP16(WT)- or USP16 (C205S) were either left uninduced (-Tet) or induced (+Tet) with tetracycline (0.5 μg/ml) for 24 hr. Cell extracts were analyzed by immunoblotting using the indicated antibodies. Note that monoubiquitinated RPS27a sometimes runs as a double band in SDS-PAGE.

The online version of this article includes the following figure supplement(s) for figure 4:

**Figure supplement 1.** Validation of USP16 knockout cell lines generated by CRISPR/Cas9.

---

inactive USP16 acts in a dominant-negative manner on removal of the RPS27a modification. The level of modified RPS27a was even increased upon depletion of endogenous USP16.

To further confirm the RPS27a modification observed upon USP16 depletion, we generated *USP16* knockout (KO) cell lines using the CRISPR/Cas9 system in HeLa and HEK293 cell lines using two different guide RNAs targeting either exon 3 (KO #1) or exon 5 (KO #2) of *USP16*. The successful knockout of *USP16* was confirmed both by immunoblotting (*Figure 4F*) and genotyping (*Figure 4*, Supporting *Figure 1*). *USP16* KO in both HeLa and HEK293 cell lines recapitulated the increase in RPS27a modification observed upon USP16 depletion (*Figure 4F*). Importantly, tetracycline-induced expression of WT but not mutant USP16-StHA in the HEK293 *USP16* KO background could rescue the *USP16* KO and led to a decrease of modified RPS27a (*Figure 4G*).

## RPS27a is trans-ubiquitinated on lysine 113 after USP16 depletion

To investigate whether the modified band observed upon USP16 depletion or deletion corresponds to mono-ubiquitinated RPS27a, we enriched RPS27a by denaturing immunoprecipitation (IP) after USP16 depletion (*Figure 5A*). Inputs and eluates of the IP were then immunoblotted for RPS27a and Ub by simultaneous detection of fluorescently labeled secondary antibodies. This revealed an overlay of the modified RPS27a band with the Ub band, indicating that depletion of USP16 leads to an accumulation of a mono-ubiquitinated form of RPS27a (*Figure 5A*). Whereas poly-ubiquitination of target proteins is often connected with subsequent proteasomal degradation, mono-ubiquitination is thought to have a non-proteolytic, regulatory effect (reviewed in *Komander et al., 2009*). Consistently, inhibition of the proteasome with MG132 leads to an increase of poly-ubiquitinated target proteins, but causes loss of regulatory mono-ubiquitination (*Higgins et al., 2015*; *Kim et al., 2011*). Treating *USP16* KO cells with MG132 for 2 or 4 hr indeed led to a strong decrease of modified RPS27a (*Figure 5B*), hinting at a regulatory role of RPS27a mono-ubiquitination.

RPS27a is produced as a linear fusion with Ub at its N-terminus and represents one of four Ub precursor proteins in human cells (reviewed in *Komander et al., 2009*). Structural analysis of eukaryotic ribosomes suggests that the Ub moiety linearly fused to RPS27a is not present in mature ribosomes and has to be removed from the precursor protein before ribosomes reach translation competence (*Rabl et al., 2011*). It has even been proposed that Ub removal occurs during or rapidly after RPS27a translation, and the DUBs UCHL3, USP7 and USP9X have been suggested to be responsible for precursor processing (*Grou et al., 2015*; *Lacombe et al., 2009*; *Monia et al., 1989*). Together with our observation that treatment of *USP16* KO cells with MG132 leads to a rapid decrease in the levels of ubiquitinated RPS27a, we hypothesized that the observed ubiquitination of RPS27a does not correspond to the linear precursor fusion protein, but to ubiquitination of RPS27a on an internal lysine (Lys) in *trans*.

High-resolution structures of 40S subunits from different eukaryotic organisms imply that several Lys residues within RPS27a coordinate to the 18S rRNA backbone (*Rabl et al., 2011*; *Voorhees et al., 2014*; *Khatter et al., 2015*) and are therefore probably not accessible for ubiquitination. Seven Lys residues (K89, K90, K96, K99, K107, K113, K152) are, however, solvent-exposed

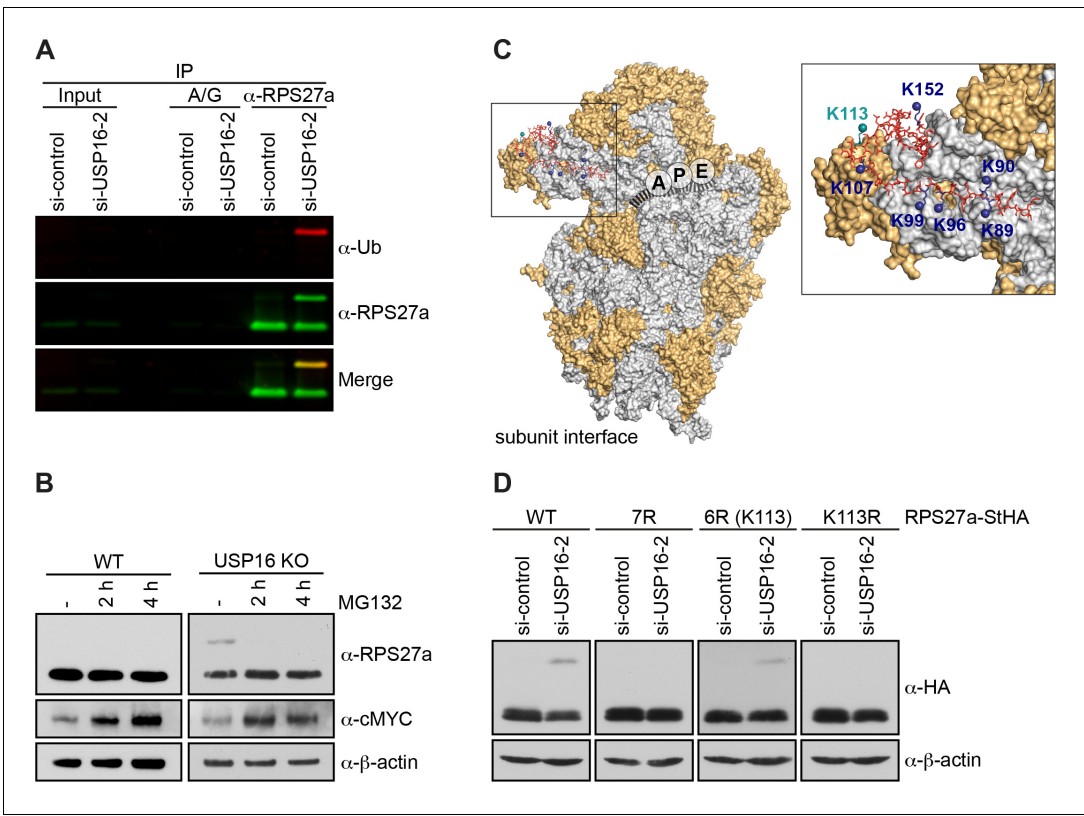

**Figure 5.** RPS27a is *trans*-ubiquitinated on lysine 113. (**A**) Denaturing immunoprecipitation (IP) of RPS27a from lysates of HeLa cells treated with control or USP16 siRNA for 72 hr. Protein A/G beads without antibody were used as negative control. Eluates were analyzed by immunoblotting using RPS27a and ubiquitin (Ub) antibodies and simultaneous detection of fluorescently labeled rabbit (for USP16) or mouse (for Ub) secondary antibodies using an Odyssey (LI-COR) imaging system. Load corresponds to 0.1% of the input and to 20% of the eluates. (**B**) HeLa WT and *USP16* KO cells were treated with MG132 (20 μM) for the indicated times and cell extracts were analyzed by immunoblotting using the indicated antibodies. (**C**) Structure of the human 40S subunit shown from the subunit interface side (adapted from *Khatter et al., 2015*, PDB ID: 4UG0), highlighting the mRNA path (dashed line), the positions of the ribosomal A, P, and E sites, and RPS27a at the 40S beak. The surfaces of RPs and rRNA are depicted in orange and light gray, respectively. RPS27a is depicted as red sticks. Mutated lysine residues are indicated in blue and teal (K113). In the inset, the positions of the mutated lysine residues are highlighted. (**D**) HeLa FlpIn cell lines expressing C-terminally StHA-tagged RPS27a WT and mutants (7R: K89/90/96/99/107/113/152R; 6R(K113): K89/90/96/99/107/152R; K113R) were treated with control or USP16 siRNA for 72 hr. Extracts were analyzed by immunoblotting using the indicated antibodies.

The online version of this article includes the following figure supplement(s) for figure 5:

**Figure supplement 1.** RPS27a(K113R)-StHA is incorporated into ribosomes.

and not directly involved in rRNA binding (*Figure 5C*). Therefore, we mutated these amino acids to arginine (Arg, R) to test whether this would abolish RPS27a ubiquitination upon USP16 depletion (*Figure 5D*). Indeed, when we depleted USP16 in cell lines expressing a C-terminally StHA-tagged RPS27a construct harboring these mutations (RPS27a(7R)-StHA), ubiquitination was no longer observed, whereas the WT construct was still modified (*Figure 5D*). To pinpoint the exact modification site, we individually reverted the mutated residues to Lys and observed that ubiquitination was reestablished when Lys113 was re-introduced (RPS27a(6R(K113))-StHA). We confirmed Lys113 as the sole modification site by its mutation to Arg (Rps27a(K113R)-StHA), which indeed abolished RPS27a ubiquitination upon USP16 depletion. This is not explained by a failure of ribosome incorporation of the RPS27a(K113R)-StHA construct, which co-sediments like the WT construct in sucrose gradients (*Figure 5*, Supporting *Figure 1*). Together, our data show that USP16 functions as a DUB for RPS27a, *trans*-mono-ubiquitinated on Lys113.

## USP16 deletion affects late stages of 40S subunit assembly

Based on our observations that USP16 associates with cytoplasmic pre-40S subunits (*Figures 1* and *2*) and ubiquitinated RPS27a co-purifies with the USP16(C205S) mutant (*Figure 4D*), we hypothesized that USP16-mediated RPS27a deubiquitination might play a role in the late stages of 40S maturation. To test this assumption, we first analyzed pre-rRNA processing by Northern blotting using a probe that hybridizes to the 5′ region of the internal transcribed spacer 1 (5′ ITS1) in 18S rRNA precursors (*Figure 6A and B*). Interestingly, while early rRNA precursors seemed unaffected in *USP16* KO cells, we observed a striking increase of the 18S-E rRNA precursor in *USP16* KO cells indicating that *USP16* deletion affects the final step of 18S rRNA processing, which is catalyzed by the endonuclease NOB1 in the cytoplasm (*Fatica et al., 2003*; *Pertschy et al., 2009*; *Preti et al., 2013*; *Sloan et al., 2013*).

To further pinpoint which cytoplasmic steps of 40S subunit assembly are impaired in *USP16* KO cells, we next analyzed potential changes in the steady state localization of 40S trans-acting factors. Such changes can be indicative of cytoplasmic recycling defects caused by 40S maturation problems. ENP1 and DIM2 are nucle(ol)ar at steady state, but both factors accompany pre-40S subunits to the cytoplasm where they are released from maturing subunits and recycled back to the nucleus (*Wild et al., 2010*; *Wyler et al., 2011*; *Zemp et al., 2009*). While ENP1 localization was not affected in HeLa *USP16* KO cells (*Figure 6C*, upper panel), DIM2 partially relocalized to the cytoplasm, suggesting a cytoplasmic 40S maturation defect that does not affect the release of ENP1 but that of DIM2 from 40S subunits in the cytoplasm. Importantly, the expression of transiently transfected WT but not mutant USP16-StHA rescued the late 40S biogenesis defect and restored nuclear localization of DIM2 (*Figure 6C and D*). Although both ENP1 and DIM2 accompany pre-40S subunits to the cytoplasm, DIM2, but not ENP1, has been shown to be part of very late 40S pre-ribosomes isolated by affinity purification of RIOK1(kd) (*Figure 1B* and *Widmann et al., 2012*). The endonuclease NOB1 is also among the very few non-ribosomal proteins associated with late pre-40S particles isolated by RIOK1(kd). All three factors, RIOK1, DIM2, and NOB1, are released from pre-40S subunits during the final steps of cytoplasmic 40S maturation. Consequently, we also examined whether loss of USP16 affects NOB1 and RIOK1 localization in *USP16* KO cells. As NOB1 and RIOK1 are cytoplasmic at steady state, we analyzed their localization after treatment of cells with LMB, which induces their accumulation in the nucleus. Strikingly, both NOB1 and RIOK1 showed enhanced cytoplasmic localization in LMB-treated *USP16* KO cells indicating an inhibition of their release from cytoplasmic pre-40S subunits (*Figure 6C*, lower panel). For both factors, the phenotype was reverted by expression of WT but not mutant USP16-StHA (*Figure 6C and D*). Together, these data indicate that *USP16* deletion only affects very late stages of cytoplasmic pre-40S maturation, after release of ENP1 and before or during release of DIM2, NOB1, and RIOK1.

## RPS27a ubiquitination is diminished upon inhibition of translation

As the deletion of USP16, which leads to RPS27a mono-ubiquitination, induces late 40S maturation defects, we wondered whether ubiquitination is a strictly ribosome biogenesis-dependent event. Therefore, we treated HeLa WT and *USP16* KO cells with a low dose of actinomycin D to specifically inhibit RNA polymerase I and thereby block ribosome assembly. This, however, did not lead to a change in the levels of ubiquitinated RPS27a (*Figure 7*, Supporting *Figure 1*), which may indicate that (pre-)40S subunits with mono-ubiquitinated RPS27a have a low turnover rate that exceeds the duration of the actinomycin D treatment or that ubiquitination of RPS27a might not be strictly dependent on ribosome maturation, for instance by also affecting mature subunits.

In a next step, we tested whether *USP16* KO and RPS27a ubiquitination also affect translation. We first examined by sucrose gradient centrifugation from HeLa WT and *USP16* KO cells whether ubiquitinated RPS27a is exclusively found in the 40S pool that also contains pre-40S subunits, or whether it also co-sediments with polysomes (*Figure 7A*). Notably, we did not detect any unassembled RPS27a, modified or unmodified, in the free pool. Interestingly, ubiquitinated RPS27a, like its unmodified form, was found in fractions containing (pre-)40S subunits, 80S ribosomes and polysomes to the same extent, suggesting that ubiquitination of RPS27a on (pre-)40S subunits does not hinder them from associating with mRNAs and entering the translating ribosome pool.

Next, to directly examine if translation is affected, we performed polysome profile analysis from HeLa WT and *USP16* KO cells (*Figure 7B*). As expected, based on our previous observations that

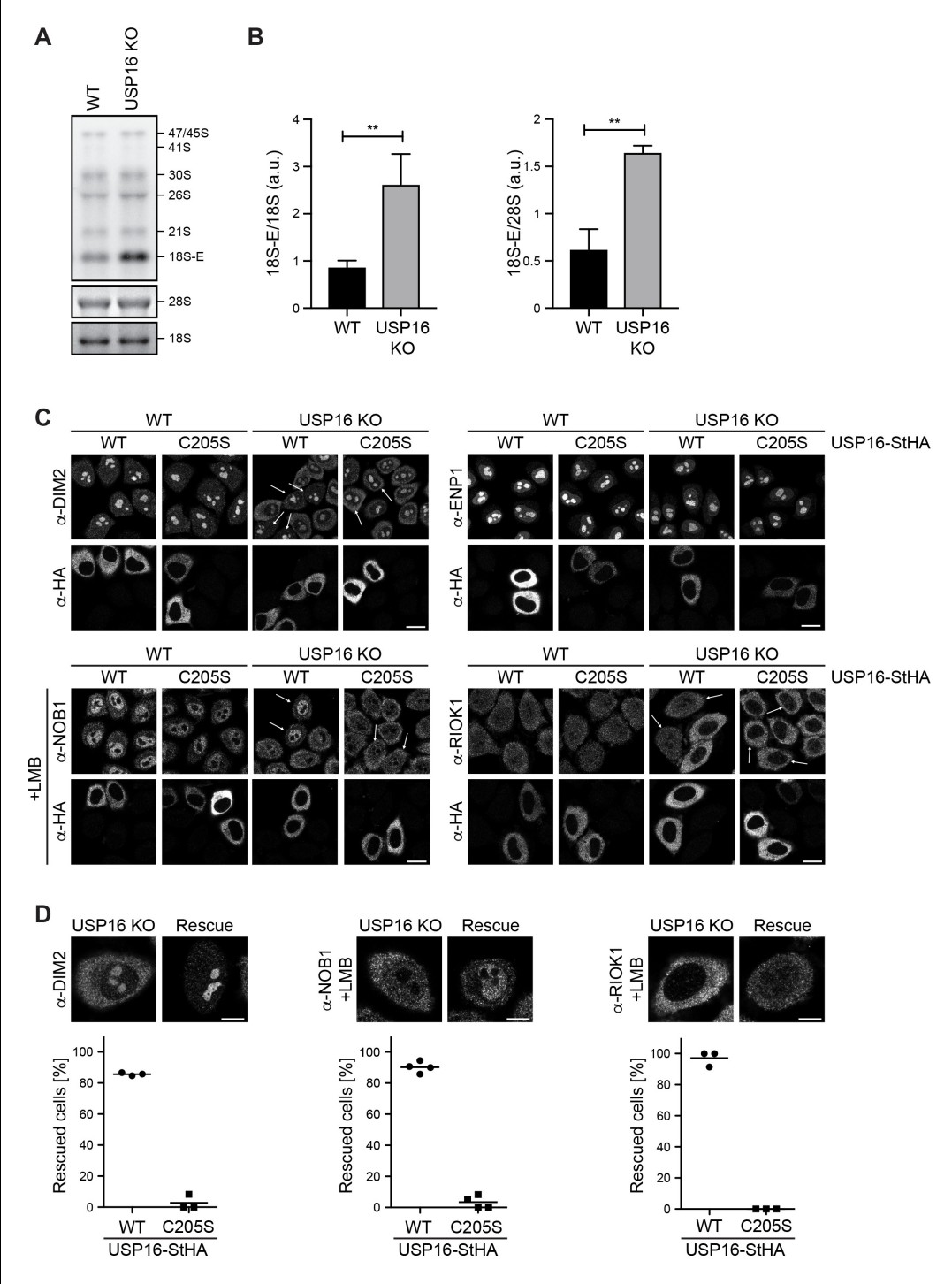

**Figure 6.** Deletion of USP16 leads to late cytoplasmic ribosome biogenesis defects. (**A**) Northern blot analysis of total RNA extracted from HeLa WT and *USP16* KO cells. A radioactively labeled 5' ITS1 probe was used to detect the indicated pre-rRNA precursors. Mature 18S and 28S rRNA were visualized by GelRed staining of the gel. (**B**) Quantification of 18S-E/18S and 18S-E/28S (pre-)rRNA levels (mean ± SD) of three independent experiments as in (**A**). **p≤0.01 (unpaired t-test). (**C**) HeLa WT and *USP16* KO cells were transfected with USP16(WT)- or USP16 (C205S)-StHA. After 24 hr, cells were analyzed by immunostaining using an antibody against the transfected construct (α-HA) and the indicated antibodies recognizing the 40S trans-acting factors ENP1, DIM2, NOB1, and RIOK1. For analyses of NOB1 and RIOK1, cells were treated with leptomycin B (LMB; 20 nM, 90 min) to inhibit CRM1-dependent nuclear export. Arrows mark cells expressing the USP16 constructs and showing phenotypic

*Figure 6 continued on next page*

*Figure 6 continued*

rescue in the *USP16* KO cell line. Scale bar, 20 μM. (**D**) Biological replicates of experiments in (**C**) were analyzed for the efficiency of rescue by transfection of USP16(WT)- or USP16(C205S)-StHA into *USP16* KO cells regarding the localization of DIM2, NOB1 and RIOK1. Example pictures of cells showing either the *USP16* KO or the rescue phenotype are shown along with the quantification of the rescue efficiency (DIM2: N = 3, n ≥ 42; NOB1: N = 4, n ≥ 67; RIOK1: N = 3, n ≥ 61). Scale bar, 10 μM.

deletion of USP16 leads to a 40S biogenesis defect, we observed an imbalance in the amount of free 40S and 60S subunits with a slight decrease in the amount of free 40S and a striking increase in the amount of free 60S subunits in *USP16* KO cells. In addition, polysome levels were slightly decreased while the 80S peak was substantially larger, which is reminiscent of changes in polysome profiles due to a translation initiation defect or different stress conditions leading to polysome disassembly and a concomitant increase of 80S monosomes (*Coudert et al., 2014*; *Kressler et al., 1997*; *Lacombe et al., 2009*; *Liu et al., 2013*). Thus, our data indicate that USP16 deletion leads to ribosomes having some defect in translation.

Previous studies have shown that pre-40S subunits can be detected in polysome fractions (*Belhabich-Baumas et al., 2017*; *Ferreira-Cerca et al., 2005*; *García-Gómez et al., 2014*; *Granneman et al., 2005*; *Lacombe et al., 2009*; *Pertschy et al., 2009*; *Soudet et al., 2010*). Whether these pre-40S particles are aberrant, enter the translation process prematurely or undergo an obligatory translation initiation-dependent quality control check remains elusive. In light of our observation that RIOK1(kd)-StHA also co-purifies several mRNA-associated proteins (*Supplementary file 2*) and that the deletion of *USP16* affects both late 40S maturation and likely protein translation (based on the changes observed in the polysome profiles), we were wondering whether RPS27a ubiquitination also depends on translation. Therefore, we investigated whether the inhibition of translation leads to changes in the ubiquitination of RPS27a (*Figure 7C*). WT and *USP16* KO cells were treated with the translation elongation inhibitor cycloheximide (CHX), the translation initiation inhibitor silvestrol (*Bordeleau et al., 2008*), or DTT, which leads to induction of the unfolded protein response and thereby to a general attenuation of translation initiation via phosphorylation of eIF2α, and were then analyzed by immunoblotting (*Figure 7C*). Whereas no changes were observed after treatment of WT cells, the levels of ubiquitinated RPS27a in *USP16* KO cells were significantly decreased after treatment with CHX or upon inhibition of translation initiation by silvestrol or DTT, suggesting that RPS27a ubiquitination is dependent on active translation. In these experiments, we also tested the aminoglycoside geneticin/G418. G418 binds to the decoding center of the ribosome and induces A-site miscoding thereby decreasing translational fidelity and promoting missense errors during protein synthesis including premature termination codon readthrough (*Garreau de Loubresse et al., 2014*; *Howard et al., 1996*; *Manuvakhova et al., 2000*; *Prokhorova et al., 2017*). Interestingly, when we treated cells with G418, we observed a significant increase in the level of ubiquitinated RPS27a in HeLa *USP16* KO cells (*Figure 7D and E*). This suggests that binding of G418 to the decoding site either induces a translational state of the ribosome that is prone for ubiquitination of RPS27a or that it stabilizes an already ubiquitinated (pre-)40S subunit. Further studies will be required to elucidate the mechanism behind this observation.

In summary, our data show that *USP16* KO and RPS27a ubiquitination not only affect late stages of 40S maturation but also translation. Furthermore, RPS27a ubiquitination is altered by small molecules that affect translation, suggesting that ubiquitination only occurs after (pre-)40S subunits have associated with mRNAs and entered the pool of translating ribosomes.

## Discussion

### USP16 is associated with cytoplasmic (pre-)40S particles

AP-MS (affinity purification-mass spectrometry) has been successfully used in the past for a first compositional characterization of human 40S pre-ribosomal subunits, which led to the identification of the most stoichiometric pre-40S interaction partners (*Widmann et al., 2012*; *Wyler et al., 2011*; *Zemp et al., 2014*). Here, we advanced this approach and performed a comprehensive characterization of the interactome of a late pre-40S particle isolated by affinity purification of a kinase-dead

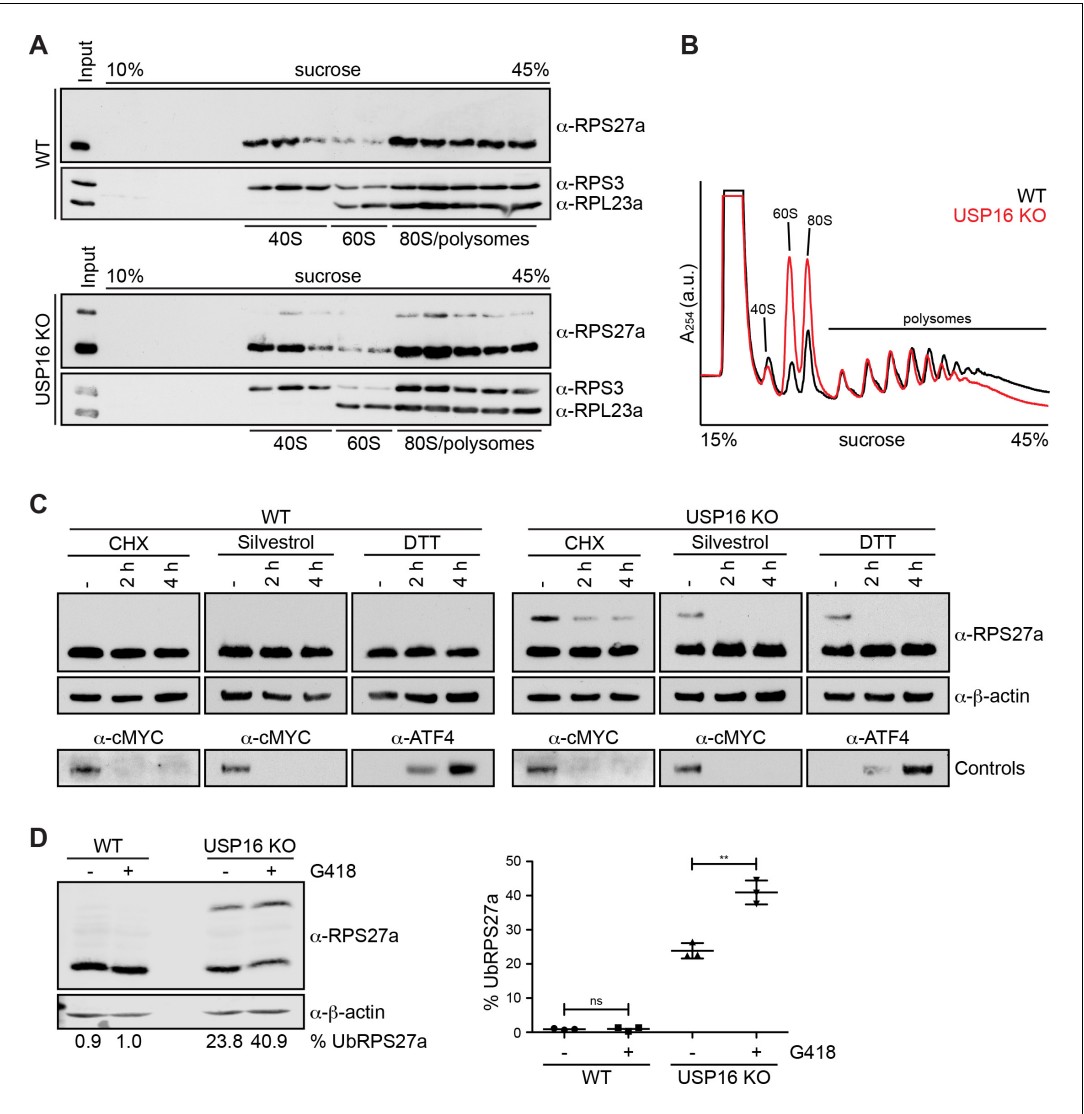

**Figure 7.** RPS27a ubiquitination is altered by interference with mRNA translation. (**A**) Extracts from HEK293 WT and *USP16* KO cells were separated on a linear 10–45% sucrose gradient by centrifugation. Input and gradient fractions were analyzed by immunoblotting using the indicated antibodies. (**B**) Extracts from HeLa WT (black line) and *USP16* KO (red line) cells were separated on a linear 10–45% sucrose gradient by centrifugation and analyzed by polysome profiling. (**C**) HeLa WT and *USP16* KO cells were treated with cycloheximide (CHX; 100 μg/ml), silvestrol (1 μM) or DTT (5 mM) for the specified times and cell extracts were analyzed by immunoblotting using the indicated antibodies. Efficiency of the treatments was assessed by immunoblotting showing rapid loss of short-lived cMYC after CHX and silvestrol treatments, and accumulation of the UPR-induced transcription factor ATF4 after DTT treatment. (**D**) HeLa WT and *USP16* KO cells were treated with G418 (400 μg/ml) for 24 hr and cell extracts were analyzed by immunoblotting using the indicated antibodies (left panel). The signal of three independent experiments was quantified and the percentage of ubiquitinated RPS27a (% UbRPS27a) of total RPS27a (modified plus unmodified RPS27a) was determined using an Odyssey (LI-COR) imaging system (mean ± SD; right panel). Values below the left panel indicate the mean of three independent experiments. **p≤0.01 (unpaired t-test).

The online version of this article includes the following figure supplement(s) for figure 7:

**Figure supplement 1.** Actinomycin D treatment does not affect RPS27a ubiquitination.
**Figure supplement 2.** Depletion of the E3 ligase ZNF598 does not affect RPS27a ubiquitination.

version of RIOK1 followed by shotgun mass spectrometry. We thereby identified a suite of novel interaction partners of RIOK1 including the deubiquitinase USP16, which we show to present a novel component of late cytoplasmic pre-40S particles. Originally, USP16 (also called Ubp-M) was identified as a DUB for histone H2A by biochemical fractionation of HeLa cell extracts (*Cai et al., 1999*; *Joo et al., 2007*). Mono-ubiquitination of H2A can serve as a repressive histone mark when deposited on chromatin by the polycomb repressive complex 1 (PRC1) through its E3 ubiquitin ligase component RNF2, or it can promote DNA damage response pathways after ubiquitination by BRCA1/BARD1 during homologous recombination or RNF168 during non-homologous end joining (reviewed in *Higashi et al., 2010*; *Uckelmann and Sixma, 2017*). H2A ubiquitination is highly dynamic and known to be counteracted by several other DUBs in addition to USP16, indicating a high level of redundancy (reviewed in *Belle and Nijnik, 2014*). Notably though, *Usp16* is essential for embryonic development of mice (*Yang et al., 2014*). However, mouse embryonic and hematopoietic stem cells bearing *Usp16* deletions are viable but show differentiation defects, indicating that USP16 might have distinct functions depending on differentiation stage (*Yang et al., 2014*).

This described chromosome-associated function of USP16 in H2A deubiquitination seems to stand in contrast to a ribosome-associated, cytoplasmic function that we have unraveled. Although USP16 has been shown to localize to chromosomes in mitotic cells and catalytically inactive USP16 has been reported to accumulate in the nucleus of interphase cells after prolonged expression (*Cai et al., 1999*; *Zhang et al., 2014*), we and others have observed that USP16 localizes primarily to the cytoplasm of cultured human somatic cells (*Cai et al., 1999*; *Sen Nkwe et al., 2020*; *Xu et al., 2013*; *Zhang et al., 2014*; *Zhuo et al., 2015*). In support of USP16 acting primarily in the cytoplasm, we found USP16 to co-purify late pre-40S particles. It is, however, possible that a minor nuclear pool acts on H2A or that, alternatively, H2A deubiquitination by USP16 mainly occurs in the course of open mitosis.

## USP16 de(p)letion leads to RPS27a ubiquitination and 40S biogenesis defects

Protein ubiquitination serves a plethora of cellular functions. Based on proteomic studies, it is estimated that a large part of the proteome is subject to ubiquitination during its lifetime, indicating that the ubiquitin system is involved in the regulation of most cellular pathways (*Clague et al., 2015*; *Kim et al., 2011*; *Swatek and Komander, 2016*). While in many cases poly-ubiquitination has been shown to induce subsequent proteasomal degradation, mono-ubiquitination has been associated with non-proteolytic, regulatory functions (*Komander and Rape, 2012*). When searching for ribosome-associated ubiquitination events that are reverted by USP16, we discovered that USP16 drives the deubiquitination of the 40S ribosomal protein RPS27a. RNAi-mediated depletion of USP16 or its CRISPR/Cas9-mediated KO in different cell lines led to RPS27a mono-ubiquitination on Lys113, supporting a ribosome-associated function of this DUB. Based on our current data, we can, however, not formally exclude that USP16 might also deubiquitinate other RPs or ribosome-bound factors. Our further analyses suggest that USP16-mediated RPS27a deubiquitination might play a role in the last stages of cytoplasmic pre-40S maturation. *USP16* KO cell lines display late 40S biogenesis defects associated with defective 18S-E pre-rRNA processing as well as mislocalization of DIM2, NOB1 and RIOK1 but not ENP1. This indicates that only events occurring after ENP1 dissociation from the maturing subunit are affected, including the timely release of DIM2, NOB1 and RIOK1.

Lys113 is situated at the ridge of the 40S beak (*Figure 4*) and is therefore highly accessible. It lies at a considerable distance to the final endonucleolytic processing site of the 18S-E pre-rRNA as well as to DIM2, NOB1 and RIOK1 that are positioned at the 40S platform and the future decoding center, respectively (*Ameismeier et al., 2018*; *Heuer et al., 2017*; *Larburu et al., 2016*; *Mitterer et al., 2019*; *Scaiola et al., 2018*). This suggests that if there were a crosstalk between the ribosome maturation machinery and the ubiquitination site, it would be indirect, either by a bridging factor or a conformational determinant.

It is currently unclear what triggers RPS27a ubiquitination and which is the responsible enzyme. One candidate is the E3 ubiquitin ligase ZNF598 which has been shown to be responsible for mono-ubiquitination of several RPs of the small subunit in the course of ribosome-associated quality control (*Garzia et al., 2017*; *Higgins et al., 2015*; *Juszkiewicz et al., 2018*; *Juszkiewicz and Hegde, 2017*; *Matsuo et al., 2017*; *Sundaramoorthy et al., 2017*). Notably, in contrast to most of the other described regulatory mono-ubiquitination sites on the 40S subunit that reside in protruding flexible

tails of ribosomal proteins, Lys113 resides in a part of RPS27a that is closely annealed to the surface of the 40S subunit. In fact, co-depletion of ZNF598 with USP16 did not induce changes in the ubiquitination levels of RPS27a (*Figure 7*, Supporting *Figure 2*), indicating that ZNF598 may not be the responsible E3 ligase. The involvement of further E3 ligases will therefore need to be assessed in the future.

## RPS27a ubiquitination – part of a quality control step of final 40S maturation?

It remains to be deciphered whether ubiquitination of RPS27a is an obligate step in 40S subunit biogenesis or part of a quality control mechanism that only affects a certain pool of subunits. The fact that only about a fifth of the RPS27a pool is mono-ubiquitinated in *USP16* KO cells could indicate that the modification is restricted to a subset of (pre-)40S particles. In turn, we currently assume that RPS27a ubiquitination must not be a prerequisite for 40S subunit maturation. Our preliminary data indeed suggest that expression of an RPS27a(K113R)-StHA construct does not induce defects in late 40S maturation (data not shown). Interestingly, while we do not observe mono-ubiquitinated RPS27a in WT cells by immunoblotting, analysis of the ubiquitin-modified proteome has identified Lys113 of RPS27a as a ubiquitination site in HCT116 cells (*Kim et al., 2011*). This could indicate that there always exists a small population of (pre-)40S subunits ubiquitinated on RPS27a. We speculate that this pool of ubiquitinated subunits might correspond to aberrant particles failing a quality control step, resulting in their ubiquitination. If USP16 is de(p)leted, such aberrant and ubiquitinated 40S precursors containing a set of 40S assembly factors would accumulate, manifesting as a late 40S biogenesis defect. RPS27a(K113R)-containing pre-40S particles might escape the quality control system and expression of this construct would therefore not lead to observable defects in 40S maturation.

Lys113 will not be obscured by binding of the 60S subunit and should not interfere with ongoing translation as no translation factors have so far been mapped to this area of the ribosome. Mono-ubiquitinated RPS27a is indeed found in fractions of sucrose gradients containing (pre-)40S subunits, 80S ribosomes and polysomes, whereas USP16 is only enriched in the (pre-)40S peak. Thus, USP16 might only be able to bind and deubiquitinate free (pre-)40S subunits while ubiquitinated (pre-)40S particles can enter the pool of translating ribosomes. As we have speculated that the ubiquitinated 40S pool could correspond to aberrant (pre-)40S particles, it is possible that these subunits can start translation initiation but fail to complete it or cannot engage in translation elongation leading to their sequestration into an inactive 80S ribosome. In support of such a hypothesis, deletion of *USP16* leads to a slight decrease in polysome levels and a striking increase of the 80S peak, which is commonly interpreted as a translation initiation defect (e.g. *Coudert et al., 2014*; *Kressler et al., 1997*; *Lacombe et al., 2009*).

RPS27a mono-ubiquitination is strongly and rapidly diminished upon inhibition of translation in *USP16* knockout cells. This observation reveals that the modification of RPS27a is highly dynamic and can even be swiftly eliminated in the absence of USP16, suggesting either the existence of DUBs acting redundantly to USP16 or degradation of modified RPS27a, for example by ribophagy of marked subunits. The loss of RPS27a mono-ubiquitination upon inhibition of translation initiation also suggests that the modification may occur only after mRNA association. We therefore propose that a fraction of pre-40S particles that are either aberrant, delayed in their maturation, or stochastically chosen from the 40S pool undergo pre-mature translation initiation, but do so inefficiently due to the presence of some late 40S trans-acting factors like DIM2, NOB1, and RIOK1. This would then initiate a quality control step leading to their mono-ubiquitination on RPS27a and possibly their dissociation from mRNAs. In the presence of USP16, these pre-40S subunits will undergo deubiquitination and final maturation or eventually degradation. In the absence of USP16, however, lack of deubiquitination would lead to an accumulation of aberrant and ubiquitinated pre-40S particles in the cytoplasm. Clearly, such a quality control step could also affect 'mature' subunits that have lost all trans-acting factors but fail to complete initiation, for instance at the start codon during formation of the first peptide bond.

In *S. cerevisiae*, a quality control step involving the formation of an 80S-like particle to proofread pre-40S particles has already been postulated (*Karbstein, 2013*; *Lebaron et al., 2012*; *Strunk et al., 2012*), but such a surveillance step has not been described in human cells. Interestingly, while Lys113 of RPS27a is conserved in *S. cerevisiae* but not *S. pombe*, USP16 and also its most closely related DUB USP45 are not conserved in yeast (*Clague et al., 2019*). Higher eukaryotes

might therefore have evolved other or additional means to control the transition from 40S precursors to mature subunits. RPS27a ubiquitination and its USP16-mediated deubiquitination could provide a system to surveil the entry of maturing pre-40S particles into the pool of translating ribosomes. Further studies will, however, be required to elucidate the molecular mechanism and consequences of RPS27a ubiquitination. On that way, it will be instrumental to identify the factors that induce as well as recognize the modification and relay the signal to downstream events. Given the essential function of USP16 at the organismal level and its various links to human diseases, we anticipate that its newly discovered role in ribosome biology will inform new hypotheses as to how the enzyme contributes to organismal homeostasis.

## Materials and methods

A list of materials is provided in *Supplementary file 1* (Key Resources Table).

### Molecular cloning

The coding sequence of USP16 was amplified from the human ORFeome collection (hORFeome V5.1, CCSB) and cloned into the pcDNA5/FRT/TO/cStHA vector encoding for a C-terminal StHA tag (St, Strep-tag II; HA, hemagglutinin epitope; described in *Wyler et al., 2011*) using the BamHI and XhoI restriction sites. USP16 truncations were subcloned from the USP16-StHA construct. The coding sequence of RPS27a was amplified from HeLa cDNA and cloned into the pcDNA5/FRT/TO/cStHA vector using the KpnI and BamHI restriction sites and into the pQE30/N-His vector using the BamHI and HindIII restriction site. The coding sequence of the RPS27a(7R) mutant construct was synthesized (Thermo Fischer Scientific) and cloned into the pcDNA5/FRT/TO/cStHA vector using the KpnI and BamHI restriction sites. The coding sequence of RPS2 was amplified from HeLa cDNA and cloned into the pQE30/N-His vector using the BamHI and HindIII restriction sites. Point mutations and deletions of USP16 and RPS27a constructs were introduced using a QuikChange mutagenesis kit (Agilent Technologies).

### Cell lines, antibodies, and reagents

HEK293 FlpIn T-REx cell lines expressing HASt-GFP, RPS2-StHA, ENP1-StHA, HASt-DIM2, HASt-LTV1, RIOK1-StHA and RIOK1(D324A)-StHA have been described previously (*Larburu et al., 2016*; *Widmann et al., 2012*; *Wyler et al., 2011*). Polyclonal HEK293 FlpIn T-REx cell lines expressing USP16-StHA were generated as described previously (*Wyler et al., 2011*). HeLa FlpIn T-REx cell lines expressing RPS27a-StHA constructs were generated as described previously (*Badertscher et al., 2015*). Cell lines used in this study were not further authenticated after obtaining them from the indicated sources. All cell lines were tested negative for mycoplasma using PCR-based testing. None of the cell lines used in this study were included in the list of commonly misidentified cell lines maintained by International Cell Line Authentication Committee.

The anti-RPS2 and anti-RPS27a antibodies were raised against purified, recombinant His-RPS2 and His-RPS27a(77-154), respectively, and affinity purified with the antigen coupled to SulfoLink beads (Thermo Fisher Scientific). Antibodies against RLP24, RPL23a, DIM2, ENP1, LTV1, NOB1, RPS3, RPS3a, NOC4L, RIOK1, RIOK2, RRP12, and TSR1 have been described previously (*Widmann et al., 2012*; *Wild et al., 2010*; *Wyler et al., 2011*; *Zemp et al., 2014*; *Zemp et al., 2009*). Anti-β-actin (A1987) was purchased from Sigma Aldrich, anti-RPL5 (ab86863), anti-RPS10 (ab151550), anti-RPS20 (ab133776), and anti-ZNF598 (ab80458) from Abcam, anti-HA (MMS-101P) from Covance, anti-ubiquitin (P4D1) (sc-8017) and anti-cMYC (sc-40) from Santa Cruz Biotechnologies, anti-ATF4 (11815) from Cell Signaling Technologies, and anti-USP16 (A301-615A) from Bethyl Laboratories.

Leptomycin B (LMB) was purchased from LC Laboratories, Silvestrol from MedChemExpress, DTT from Applichem, G418 from Thermo Fisher Scientific, and cycloheximide (CHX), N-ethylmaleimide (NEM) and MG132 were purchased from Sigma Aldrich.

### Immunoblot analysis

For immunoblot analysis, whole cell extracts in SDS sample buffer were separated on SDS-PAGE gels and proteins were transferred to nitrocellulose membranes by semi-dry blotting. For immunoblot analysis using the α-ubiquitin (P4D1) antibody, proteins were transferred to a PVDF membrane

before the membrane was denatured in 6 M guanidinium chloride, 20 mM Tris pH 7.5, 100 µM PMSF, 5 mM β-mercaptoethanol. Membranes were then blocked in 4% milk in PBST and incubated with primary antibodies, washed in PBST and incubated with secondary antibodies. Signals were captured by exposure to film or detected using using a Fusion (Vilber) or an Odyssey (LI-COR) imaging system.

## Transient transfection and RNA interference

Transient transfection of plasmid DNA was performed using X-tremeGENE 9 DNA transfection reagent (Roche) for 24 hr. Transfection of siRNAs into HeLa cells was carried out at a final oligonucleotide concentration of 10 nM using INTERFERin (Polyplus transfection) and into HeLa FlpIn and HEK293 cells at a final oligonucleotide concentration of 15 or 20 nM, respectively, using Lipofectamine RNAiMax (Invitrogen).

The following siRNA oligonucleotides were used in this study:

Allstars siRNA (Qiagen) (negative control; si-control), si-USP16-2 (5'-AAUGGCUGAAAUAACGA UAAA-3'), si-USP16-3 (5'-CCUCCUGUUCUUACUCUUCAUUUAA-3'), si-ZNF598-1 (5'-CAGGACUAC UACAGCGACUAU-3'), si-ZNF598-2 (5'-ACAAAUGGUCCUGUAAGCCAA-3'), si-ZNF598-3 (5'-UGGAAAGGUGUACGCAUUGUA-3'), si-ZNF598-4 (5'- CACAGAUGUGUUGUGUAAACA-3').

## Generation of knockout cell lines

*USP16* KO cell lines were generated using the CRISPR/Cas9 system. Guide RNAs (gRNAs) were predicted using a CRISPR design web tool (http://crispr.mit.edu). gRNA target sites were within exon 3 (USP16 gRNA 1: 5'-CACCGTATTGTCAGTCTTACAGTCT-3' and 5'-AAACAGACTGTAAGACTGA-CAATAC) or exon 5 (USP16 gRNA 2: 5'-CACCGAATCAACCACTTGACCCAAC-3' and 5'-AAACG TTGGGTCAAGTGGTTGATTC-3'). Annealed gRNAs were ligated into the pC2P vector (*Welte et al., 2019*), which encodes hCas9 and contains a puromycin resistance cassette. For generation of KO cell lines, cells were transfected with pC2P vector containing the gRNA sequence and selected with puromycin for 3 days. Individual clones were expanded and screened for mutations in the USP16 gene by PCR and immunoblotting. PCR products were sequenced and analyzed for indel mutations using the tide web tool (http://tide.nki.nl *Brinkman et al., 2014*) and by manual inspection of the sequencing profiles.

## StrepTactin pull-downs and tandem affinity purification

Preparation of cell extracts for StrepTactin pull-downs and tandem affinity purifications (TAP) was performed as described previously (*Wyler et al., 2011*). In short, expression of the bait protein in HEK293 FlpIn T-REx cell lines was induced with tetracycline (0.5 µg/ml) for 24 hr. Then, cells were detached with PBS containing 0.5 mM EDTA and harvested by centrifugation (900 g, 5 min, 4˚C). Cells were lysed in 10 mM Tris pH 7.5, 100 mM KCl, 2 mM MgCl$_2$, 0.5% NP-40, 1 mM DTT containing protease and phosphatase inhibitors using a dounce homogenizer and lysates were cleared by centrifugation (4500 g, 12 min, 4˚C). For StrepTactin pull-downs, extracts were incubated with StrepTactin beads (IBA) for 30 min at 4˚C while rotating. Beads were washed three times with TAP buffer (10 mM Tris pH 7.5, 100 mM KCl, 2 mM MgCl$_2$ containing protease and phosphatase inhibitors) and once with 10 mM Tris pH 7.5, 2 mM MgCl$_2$ before bound proteins were eluted with sample buffer without DTT before supplementation with 50 mM DTT.

For mass spectrometry analysis of complexes purified by StrepTactin affinity purification, extracts were essentially prepared as described above, but the lysis buffer was additionally supplemented with 2 µM avidin. Instead of elution with sample buffer, proteins were eluted with TAP buffer containing 2.5 mM *d*-desthiobiotin (Sigma). After precipitation with trichloroacetic acid (TCA), proteins were denatured with 6 M urea, reduced with 12 mM DTT and alkylated with 40 mM iodoacetamide. Samples were diluted to 1 M urea with 0.1 M NH$_4$HCO$_3$ before addition of trypsin (Promega) and incubation O/N at 30˚C. The digest was stopped by addition of 2% formic acid and peptides were cleaned up using Pierce C-18 spin columns (Thermo Fisher Scientific) according to the manufacturer's protocol.

For TAP, extracts were prepared as described above and were incubated with StrepTactin beads (IBA) for 30 min at 4˚C while rotating. Beads were washed three times with TAP buffer and eluted three times with TAP buffer containing 2.5 mM *d*-desthiobiotin (Sigma). Eluates were pooled and

incubated with HA-agarose (Sigma) for 1 hr at 4°C while rotating. Beads were washed three times with TAP buffer (10 mM Tris pH 7.5, 100 mM KCl, 2 mM MgCl$_2$ containing protease and phosphatase inhibitors) and once with 10 mM Tris pH 7.5, 2 mM MgCl$_2$. Bound proteins were eluted with SDS sample buffer without DTT before supplementation with 50 mM DTT.

### Immunofluorescence analysis

Immunofluorescence analysis was performed as described previously (*Zemp et al., 2009*). Images were acquired using a Leica SP2 AOBS microscope using a 63 × 1.4 NA, oil, HCX Plan-Apochromat objective or a Zeiss LSM880 upright microscope with a 63 × 1.4 NA, oil, DIC Plan-Apochromat objective.

### Sucrose gradients

HeLa cells were treated with 100 µg/ml cycloheximide for 3 min and lysed in 10 mM Tris pH 7.5, 100 mM KCl, 10 mM MgCl$_2$, 1% Triton X-100, 1 mM DTT, 100 µg/ml cycloheximide, and protease inhibitors. The lysate was centrifuged (10'000 g for 3 min at 4°C) and the resulting supernatant was used for sucrose gradient analysis. For polysome profiling analysis, extracts (1.8 mg of total protein) were loaded onto a linear 15–45% (w/v) sucrose gradient in 50 mM HEPES-KOH pH 7.5, 100 mM KCl, 10 mM MgCl$_2$. After centrifugation for 225 min at 26'800 rpm at 4°C in a SW41 rotor (Beckman Coulter), gradients were analyzed at OD$_{254}$ with a Foxy Jr. Gradient collector (Teledyne Isco). For analysis of sucrose gradients by immunoblotting, extracts (400–800 µg of total protein) were loaded onto a linear 10–45% (w/v) sucrose gradient in 50 mM HEPES-KOH pH 7.5, 100 mM KCl, 3 mM MgCl$_2$. After centrifugation for 80 min at 55'000 rpm at 4°C in a TLS55 rotor (Beckman Coulter), fractions were precipitated with trichloroacetic acid (TCA) and used for immunoblot analysis.

### Denaturing immunoprecipitation

HeLa cells were washed briefly with PBS, detached with PBS containing 0.5 mM EDTA and 2 mM NEM and centrifuged (800 g, 5 min, 4°C). Cell pellets were resuspended in denaturing lysis buffer (1% SDS, 5 mM EDTA, 10 mM DTT, 15 U/ml DNase I and protease inhibitors), vortexed vigorously and denatured for 5 min at 95°C. After centrifugation (16'000 g, 1 min, RT), the supernatant was diluted 1:10 with wash buffer (10 mM Tris pH 7.5, 1 mM EDTA, 1 mM EGTA, 150 mM NaCl, 1% Triton X-100, 20 mM NEM and protease inhibitors) and mixed gently before passage through a 27G needle, incubation on ice for 5 min and centrifugation (16'000 g, 10 min, 4°C). The supernatant was incubated with or without RPS27a antibody for 1 hr at 4°C while rotating and added to A/G beads equilibrated in wash buffer. After incubation for 2 hr at 4°C on a rotating wheel, beads were washed twice in wash buffer and once in 10 mM Tris pH 7.5, 2 mM MgCl$_2$ and bound proteins were eluted with 50 µl SDS sample buffer without DTT before supplementation with 50 mM DTT.

### Northern blot analysis

Total RNA was extracted from HeLa cells using the RNeasy Mini kit (QIAGEN). Northern blot analysis was performed as described previously (*Tafforeau et al., 2013*). In short, 2 µg RNA were separated on an agarose-formaldehyde gel (75 V, 4.5 hr) and stained with GelRed (Biotium). RNA was then transferred to a nylon membrane (Hybond-N$^+$, GE Healthcare) by capillary transfer and cross-linked by UV light. rRNA precursors were analyzed with the radioactively labeled 5' ITS1 probe (5'-CC TCGCCCTCCGGGCTCCGTTAATGATC-3', *Rouquette et al., 2005*). The membrane was analyzed by phosphor-imaging using a Typhoon FLA 9000 (GE Healthcare). Quantification was performed using ImageJ.

### Mass spectrometric analyses

Nanoflow LC-MS/MS measurements were carried out on an EASY-nLC 1000 (Thermo Fisher) coupled to a Q Exactive Plus mass spectrometer (Thermo Fisher) equipped with a Nanospray Flex ion source. The peptides were separated on a 75 µm diameter, 15 cm-long new Objective emitter packed with ReproSil Gold 120 C18 resin (1.9 µm, Dr. Maisch) and eluted at 300 nl/min with a linear gradient of 5% to 30% Buffer A for either 90 min (dataset 1) or 120 min (dataset 2). The composition of buffers was as follows: Buffer A: 0.1% formic acid; Buffer B: 99.9% acetonitrile, 0.1% formic acid. MS data acquisition was performed in data-dependent acquisition (DDA, top20, excluding the singly and

unassigned charged species, with 30 s dynamic exclusion). The collision energy was set to 30 NCE. The resolution of the Orbitrap analyzer was set to 70,000 and 17,500 for MS1 and MS2, with a maximum injection time of 64 ms and 55 ms, respectively. The AGC target was set to 3e6 in MS1 and 1e5 in MS2. Data were acquired with Xcalibur version 4.2.28.14.

For data analysis, the DDA search was performed on mzXML files obtained from msconvert (Proteowizzard v 3.0.18304) with Comet (2018.01 rev. 2) (*Eng et al., 2013*) using a human database (downloaded from UniProt on 17.04.2019) filtered for reviewed Swissprot entries only (20'402 proteins), supplemented with one entry for the concatenated sequence of the iRT peptides (*Escher et al., 2012*) and as many decoy protein entries generated by pseudo-reversing the tryptic peptide sequences. The database was further supplemented with the protein sequences of HASt-GFP, and RIOK1 WT and mutant (dataset 1) or USP16 WT and mutant (dataset 2). The search parameters were as follows: +/- 25 ppm tolerance for MS1 and MS2, fixed cysteine carbamidomethylation, variable methionine oxidation and protein N-termini acetylation, semi-tryptic and two missed cleavages allowed. The comet search results were further processed using peptideProphet (*Keller et al., 2002*) separately for both datasets and filtered to 1% false discovery rate. The protein spectral count tables were then processed by the SAINT express algorithm (*Teo et al., 2014*) on the crapome.org website (*Mellacheruvu et al., 2013*). The spectral counts of the confidently identified interactors (SAINT Bayesian-FDR below 0.01) were then normalized to that of the bait for each dataset and, in case of RIOK1, finally tested by ANOVA to obtain the set of interactors significantly changing between the conditions (mutant vs. WT baits).

## Acknowledgements

We thank Dr. Sumit Pawar for critical reading of the manuscript, and the members of the Kutay lab for helpful discussions. We are very grateful to our NCCR colleagues Lara Contu and Prof. Oliver Mühlemann for exploring potential links to NMD. Microscopy was performed on instruments of the ETHZ Microscopy Center Scope M. This work was initially funded by a grant of the Swiss National Science Foundation (SNSF) to UK (31003A_166565), and subsequently by the NCCR 'RNA and disease'. SJ was supported by a FEBS long-term fellowship.

## Additional information

### Funding

| Funder | Grant reference number | Author |
| --- | --- | --- |
| Schweizerischer Nationalfonds zur Förderung der Wissenschaftlichen Forschung | NCCR RNA and disease | Ulrike Kutay |
| Schweizerischer Nationalfonds zur Förderung der Wissenschaftlichen Forschung | 31003A_166565 | Ulrike Kutay |
| FEBS | Fellowship | Stefanie Jonas |

The funders had no role in study design, data collection and interpretation, or the decision to submit the work for publication.

### Author contributions

Christian Montellese, Conceptualization, Data curation, Formal analysis, Investigation, Visualization, Methodology, Writing - original draft, Writing - review and editing; Jasmin van den Heuvel, Kerstin Dörner, Investigation, Visualization, Writing - review and editing; Caroline Ashiono, André Melnik, Investigation; Stefanie Jonas, Investigation, Methodology, Writing - review and editing; Ivo Zemp, Supervision, Writing - original draft, Writing - review and editing; Paola Picotti, Supervision, Methodology; Ludovic C Gillet, Data curation, Formal analysis, Investigation, Methodology; Ulrike Kutay, Conceptualization, Data curation, Formal analysis, Supervision, Funding acquisition, Writing - original draft, Project administration, Writing - review and editing

## Author ORCIDs

Ludovic C Gillet  http://orcid.org/0000-0002-1001-3265
Ulrike Kutay  https://orcid.org/0000-0002-8257-7465

## Decision letter and Author response

Decision letter https://doi.org/10.7554/eLife.54435.sa1
Author response https://doi.org/10.7554/eLife.54435.sa2

## Additional files

### Supplementary files

- Supplementary file 1. Key Resources Table.

- Supplementary file 2. Proteomic analysis of the RIOK1 wild-type and kinase-dead interactome S2-1 Spectral counts of proteins identified on HASt-GFP, RIOK1(WT)- and RIOK1(kd)-StHA in three independent biological replicates. S2-2 Spectral counts of proteins identified on RIOK1(WT)- and RIOK1(kd)-StHA after normalization to protein length (spectral counts per 1000 amino acids) and to spectral counts of the RIOK1(WT)-StHA bait in replicate II after filtering against the HASt-GFP control. S2-3 Comparison of the normalized spectral counts of proteins identified on RIOK1(WT)- and RIOK1(kd)-StHA. Proteins with an adjusted p value < 0.05 and an at least twofold enrichment on RIOK1(kd)-StHA ($\log_2$FC (kd vs. WT)>1) are shown in black, proteins with lower significance are depicted in dark gray (adjusted p value < 0.05 but with a $\log_2$FC < 1) or light gray (adjusted p value > 0.05).

- Supplementary file 3. Proteomic analysis of the interactome of USP16 wild-type and the catalytically-dead mutant. S3-1 Spectral counts of proteins identified on HASt-GFP, USP16(WT)- and USP16(C205S)-StHA in three independent biological replicates (I-III). S3-2 Spectral counts of proteins identified on RIOK1(WT)- and RIOK1(kd)-StHA after normalization. Spectral counts were normalized to protein length (spectral counts per 1000 amino acids) and to spectral counts of the USP16(WT)-StHA bait in replicate II after filtering against the HASt-GFP control.

- Transparent reporting form

### Data availability

The mass spectrometry proteomics data have been deposited to the ProteomeXchange Consortium via the PRIDE [1] partner repository with the dataset identifier PXD016458 (http://www.ebi.ac.uk/pride/archive/projects/PXD016458).

The following dataset was generated:

| Author(s) | Year | Dataset title | Dataset URL | Database and Identifier |
|---|---|---|---|---|
| Montellese C, Van denHeuvel J, Ashiono C, Dörner K, Melnik A, Jonas S, Zemp I, Picotti P, Gillet L, Kutay U | 2019 | AP-MS analysis of human RIOK1 and USP16 | http://www.ebi.ac.uk/pride/archive/projects/PXD016458 | PRIDE, PXD016458 |

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
