## [Decision Letter]

**Acceptance summary:**

This study describes a late step in the 40S ribosomal subunit maturation. The authors identified by mass spectrometry several proteins that were not previously known to be involved in ribosome maturation. One of these proteins is USP16. Although USP16 was first discovered as a deubiquitinase (DUB) that acts on histone H2A to regulate chromatin and cell cycle progression, the authors convincingly demonstrate that USP16 deubiquitinates RPS27A in a translation-dependent manner. They identified the domain that recruits it to the 40S pre-ribosomal particles, and the ubiquitination site that persists on Rps27a in the absence of USP16. It is not immediately clear by which mechanism mono-ubiquitination of RPS27a is reverted upon translational inhibition in USP16 knockout cells. Future studies should reveal this mechanism and the mechanistic role of Rsp27a ubiquitination and de-ubiquitination in 40S ribosomal subunit maturation.

**Decision letter after peer review:**

Thank you for submitting your article "USP16 counteracts mono-ubiquitination of RPS27a and promotes maturation of the 40S ribosomal subunit" for consideration by *eLife*. Your article has been reviewed by two peer reviewers, and the evaluation has been overseen by James Manley as the Senior Editor. The reviewers have opted to remain anonymous.

The reviewers have discussed the reviews with one another and the Reviewing Editor has drafted this decision to help you prepare a revised submission.

Summary:

The authors used mass spectrometry to identify USP16 as a new late 40S assembly factor critical for 40S maturation. They present strong evidence to support this conclusion. They demonstrate that USP16 deubiquitinates RPS27A in a translation-dependent manner. They showed that USP16 is associated with 40S pre-ribosomal particles, and identify the domain that recruits it. They also describe the ubiquitination site on Rps27a, which is the target of USP16. Importantly, they show that 40S biogenesis is impaired in the absence of USP16.

Essential revisions:

The paper presents novel and very convincing data. The reviewers have a few excellent suggestions to further strengthen the paper, including a more informed Discussion and removing the translation data (Figure 6), which is a separate (to be further developed) story. Please, see the reviewers' comments below:

*Reviewer #1:*This study has investigated a late step in 40S ribosomal subunit maturation. The approach was to affinity purify late maturation intermediate(s) via RIOK1. Mass spectrometry revealed several enriched proteins that were not previously known to be involved in ribosome maturation. Of these, the authors followed up on USP16. They show convincingly that USP16 is indeed associated with 40S pre-ribosomal particles, identify the domain that recruits it, identify the ubiquitination site that persists on Rps27a in the absence of USP16, and show that 40S biogenesis is impaired in the absence of USP16. In the final figure, the authors show that ubiquitination of Rps27a is influenced by translation.

The findings convincingly assign a new role for USP16 and point to a new player in 40S biogenesis. Although the mechanistic role of Rsp27a ubiquitination and de-ubiquitination in maturation remains unclear, this study provides a solid foundation from which to examine these issues. I therefore support publication of this study and have the following suggestions for improvement.

1) I'm not really sure the authors can claim that RPS27a modification is translation dependent. The observation is that they don't see the modification in USP16 knockout cells when translation is impaired. However, there might be other DUBs that act on this site under these conditions. Our knowledge of regulatory ubiquitination on the ribosome is very poor at this time, so one does not really know the range of ligases and DUBs and their relative relationships. In general, the relationship to translation is rather muddled. I wonder whether the paper would be simpler and more straightforward if it ended after Figure 6, leaving the translation aspect for a future study.

2) I feel the authors should better use the available structural information in their Discussion. I think readers will want to know, perhaps using a cartoon model, where the actual modification on RPS27a is and how this position relates to a number of important biological processes. Some important points a reader might want to know: (i) the site is solvent exposed and not obscured by the 60S or any translation factors; (ii) it is near the entrance to the mRNA channel, but does not line the channel; (iii) it is part of a structured region of RPS27a, which is different than the other regulatory modifications (e.g., by ZNF598) which are typically in unstructured tails that are not seen in ribosome structures. In addition to the above, the authors should certainly comment on where the modification is relative to the specific 18S processing reaction that seems to be impaired, or the 40S biogenesis factors that seem to be affected. This would make the paper more accessible to readers not directly in the ribosome biogenesis field.

*Reviewer #2:*To date, several late pre-40S (small) ribosomal subunit processing factors have been shown to be crucial for the final cytoplasmic pre-rRNA cleavages and maturation transitioning that generates mature 40S subunits. In this work the authors use proteomics to identify USP16 as a novel late 40S assembly factor critical for 40S maturation. Although USP16 was first discovered as a deubiquitinase (DUB) that acts on histone H2A to regulate chromatin and cell cycle progression, the authors convincingly demonstrate that USP16 deubiquitinates RPS27A in a translation-dependent manner, enabling important final steps in 40S maturation in human tissue culture cells.

First, USP16 is shown to strongly coimmunoprecipitate with late pre-40S particles containing tagged RIOK1 as a marker. Mutational analysis reveals that (1) the zinc-finger domain and a basic helix in the USP domain are major and minor contributors to pre-40S incorporation, respectively, and (2) Lys 113 is trans-ubiquitinated in vivo. RNAi and CRISPR depletion of USP16 augments levels of Ub-RPS27A, which can be rescued by expression of WT but not catalytically-dead USP16. USP16 depletion also causes a defect in late cytoplasmic pre-18S rRNA precursor processing and in proper localization of other known 40S assembly factors. On one hand, USP16 depletion causes retention of Ub-RPS27A modification and polysome degeneration, indicative of translational interruption. However, direct small molecule inhibition of translation in a USP16 KO background causes a loss of Ub-RPS27A, suggesting a subtle interplay between RPS27A modification and translational health.

Overall, these results highlight an important new role for USP16 in small ribosomal subunit biogenesis. The figures in the paper are easy to follow, and support the authors' interpretation of USP16's novel functions, on the whole. It will be interesting to see follow-up work on USP16 and the other late pre-40S MS/MS hits found in the manuscript.

---

## [Author Response]

Essential revisions:The paper presents novel and very convincing data. The reviewers have a few excellent suggestions to further strengthen the paper, including a more informed Discussion and removing the translation data (Figure 6), which is a separate (to be further developed) story. Please, see the reviewers' comments below:

We thank the reviewers and the editors for the very positive evaluation of our work and their constructive criticism. As suggested, we have revised the Discussion, putting more emphasis on structural aspects of the RPS27a modification (please see response to point 2 of reviewer #1).

We have also carefully evaluated the suggestion of removing the data on the link between RPS27a ubiquitination and translation. However, we feel that the data reveal an intriguing observation that we expect to be of great interest to researchers that work on ribosome biogenesis or ribosome-associated quality control. While we agree that this aspect of our work needs to be further developed in the future, we consider these data solid and informative for future studies in the field. Still, we have taken the editorial and reviewer's suggestion into account in that we have toned down our conclusions as well as revised the respective part of the Discussion based on the considerations of reviewer #1 (please see response to point 1 of reviewer #1). If indeed necessary, we would consider transferring the respective figure into the supplement.

Reviewer #1:[...] The findings convincingly assign a new role for USP16 and point to a new player in 40S biogenesis. Although the mechanistic role of Rsp27a ubiquitination and de-ubiquitination in maturation remains unclear, this study provides a solid foundation from which to examine these issues. I therefore support publication of this study and have the following suggestions for improvement.1) I'm not really sure the authors can claim that RPS27a modification is translation dependent. The observation is that they don't see the modification in USP16 knockout cells when translation is impaired. However, there might be other DUBs that act on this site under these conditions. Our knowledge of regulatory ubiquitination on the ribosome is very poor at this time, so one does not really know the range of ligases and DUBs and their relative relationships. In general, the relationship to translation is rather muddled. I wonder whether the paper would be simpler and more straightforward if it ended after Figure 6, leaving the translation aspect for a future study.

The reviewer indeed points at a very puzzling aspect of our translation inhibition experiments as it remains unresolved by which mechanism mono-ubiquitination of RPS27a is reverted upon translational inhibition in USP16 knockout cells. There exist several possibilities. Firstly, as indicated by the reviewer, there could be a redundant, homologous DUB that contributes to recycling of the affected subunits. Candidates include USP45, the closest homolog of USP16, which we, however, excluded as it seems not to be expressed in HeLa cells (Itzhak et al., *eLife*, 2016). Secondly, deubiquitination could be linked to a rescue of 40S subunits from degradation via USP10-mediated deubiquitination of RPS2, RPS3 and RPS10, as recently suggested by the Tuschl lab (Meyer et al., Mol. Cell, 2020). Thirdly, another recent study from the Bennett lab identified USP21 and OTUD3 as DUBs that counteract RP monoubiquitination linked to ribosome-associated quality control initiated by the E3 ligase ZNF598 (Garshott et al., *eLife*, 2020). While we exclude ZNF598 to act on RPS27a, a redundant function of the other identified DUBs remains to be tested. Thus, as commented by the reviewer, the relation of various DUBs to the ligases that they counteract is only beginning to emerge. Finally, it is also possible that subunits marked by mono-ubiquitination of RPS27a are turned over upon inhibition of translation, e.g. by ribophagy.

However, we think that our data clearly demonstrate that inhibition of translation leads to a loss of mono-ubiquitination of RPS27a. This also entails, as proposed by us, the possibility that the modification is posited on the 40S subunit by a translation-dependent mechanism in first place. As discussed above, we consider our data to be very solid and of high interest to the field, and therefore prefer keeping them in the manuscript. However, we have toned down our conclusions on the translation dependence of RPS27a throughout the manuscript, and we now also point at other options in the revised Discussion, to read:

"RPS27a mono-ubiquitination is strongly and rapidly diminished upon inhibition of translation in USP16 knockout cells. [...] The loss of RPS27a mono-ubiquitination upon inhibition of translation initiation also suggests that the modification may occur only after mRNA association".

2) I feel the authors should better use the available structural information in their Discussion. I think readers will want to know, perhaps using a cartoon model, where the actual modification on RPS27a is and how this position relates to a number of important biological processes. Some important points a reader might want to know: (i) the site is solvent exposed and not obscured by the 60S or any translation factors; (ii) it is near the entrance to the mRNA channel, but does not line the channel; (iii) it is part of a structured region of RPS27a, which is different than the other regulatory modifications (e.g., by ZNF598) which are typically in unstructured tails that are not seen in ribosome structures. In addition to the above, the authors should certainly comment on where the modification is relative to the specific 18S processing reaction that seems to be impaired, or the 40S biogenesis factors that seem to be affected. This would make the paper more accessible to readers not directly in the ribosome biogenesis field.

We fully agree with this point and would like to thank the reviewer for the suggestion. We have **revised Figures 4 and 5** to provide the reader with structural models of where RPS27a and other RPSs known to undergo regulatory mono-ubiquitination reside on the 40S subunit. Furthermore, as suggested, we have extended the Discussion, now providing some considerations regarding these structural aspects, to read:

"Lys113 is situated at the ridge of the 40S beak (Figure 4) and is thus highly accessible. It lies at a considerable distance to the final endonucleolytic processing site of the 18S-E pre-rRNA as well as to DIM2, NOB1 and RIOK1 that are positioned at the 40S platform and the future decoding center, respectively (Ameismeier et al., 2018, Heuer et al., 2017b, Larburu et al., 2016, Mitterer et al., 2019, Scaiola et al., 2018). This suggests that a potential crosstalk between the ribosome maturation machinery and the ubiquitination site might be indirect, either by a bridging factor or a conformational determinant".

"Notably, in contrast to most of the other described regulatory mono-ubiquitination sites on the 40S subunit that reside in protruding flexible tails of ribosomal proteins, Lys113 resides in a part of RPS27a that is closely annealed to the surface of the 40S subunit. In fact, co-depletion of ZNF598..."

"Lys113 will not be obscured by binding of the 60S subunit and, to our knowledge, should not interfere with ongoing translation as no translation factors have so far been mapped to this area of the ribosome. Mono-ubiquitinated RPS27a is indeed found..."